



# Climate change in the High Mountain Asia in CMIP6

Mickaël Lalande[1], Martin Ménégoz[1], Gerhard Krinner[1], Kathrin Naegeli[2], and Stefan Wunderle[2]

[1]Univ. Grenoble Alpes, CNRS, IRD, G-INP, IGE, 38000 Grenoble, France
[2]Institute of Geography and Oeschger Center for Climate Change Research, University of Bern, 3012 Bern, Switzerland

**Correspondence:** Mickaël Lalande (mickael.lalande@univ-grenoble-alpes.fr)

**Abstract.** Climate change over High Mountain Asia (HMA, including the Tibetan Plateau) is investigated over the period 1979-2014 and in future projections following the four shared socioeconomic pathways SSP1-2.6, SSP2-4.5, SSP3-7.0 and SSP5-8.5. The skill of 26 CMIP6 models is estimated for near-surface air temperature, snow cover extent and total precipitation, and 10 of them are used to describe their projections until 2100. Similarly to previous CMIP models, this new generation of GCMs shows a mean cold bias over this area reaching $-1.9$ [$-8.2$ to $2.9$] °C (90 % confidence interval) in comparison with the CRU observational dataset, associated with a snow cover mean overestimation of 12 [$-13$ to $43$] %, corresponding to a relative bias of 52 [$-53$ to $183$] % in comparison with the NOAA CDR satellite dataset. The temperature and snow cover model biases are more pronounced in winter. Simulated precipitation rates are overestimated by 1.5 [0.3 to 2.9] mm day$^{-1}$, corresponding to a relative bias of 143 [31 to 281] %, but this might be an apparent bias caused by the undercatch of solid precipitation in the APHRODITE observational reference. For most models, the cold surface bias is associated with an overestimation of snow cover extent, but this relationship does not hold for all models, suggesting that the processes of the origin of the biases can differ from one model to another one. A significant correlation between snow cover bias and surface elevation is found, and to a lesser extent between temperature bias and surface elevation, highlighting the model weaknesses at high elevation. The models performing the best for temperature are not necessarily the most skillful for the other variables, and there is no clear relationship between model resolution and model skill. This highlights the need for a better understanding of the physical processes driving the climate in this complex topographic area, as well as for further parameterization developments adapted to such areas. A dependency of the simulated past trends to the model biases is found for some variables and seasons, however, some highly biased models fall within the range of observed trends suggesting that model bias is not a robust criterion to discard models in trend analysis. The HMA median warming simulated over 2081-2100 with respect to 1995-2014 ranges from 1.9 [1.2 to 2.7] °C for SSP1-2.6 to 6.5 [4.9 to 9.0] °C for SSP5-8.5. This general warming is associated with a relative median snow cover extent decrease from $-9.4$ [$-16.4$ to $-5.0$] % to $-32.2$ [$-49.1$ to $-25.0$] % and a relative median precipitation increase from 8.5 [4.8 to 18.2] % to 24.9 [14.4 to 48.1] % by the end of the century in these respective scenarios. The warming is 11 % higher over HMA than over the other Northern Hemisphere continental surfaces excluding the Arctic area. Seasonal temperature, snow cover and precipitation changes over HMA show a linear relationship with the Global Surface Air Temperature (GSAT), except for summer snow cover that shows a slower decrease at strong levels of GSAT.



# 1 Introduction

High Mountain Asia (HMA) extends from the Himalayas in the south and east, to the Hindu Kush in the west and to the Tien Shan in the north, including also the Karakoram, the Pamir-Alay and the Kunlun mountain ranges. HMA surrounds the Tibetan Plateau (TP) which is the highest and most extensive plateau in the world, with an average elevation of 4000 m above sea level and an approximate surface area of 2.5 million $km^2$ (Du and Qingsong, 2000). Because of their high elevation and complex terrain, TP and HMA affect not only the regional climate and environment in East Asia, but also the global atmospheric circulation via thermal and mechanical forcings (Flohn, 1957; Kutzbach et al., 1993; Webster et al., 1998; Hsu and Liu, 2003; Duan and Wu, 2005; Liu et al., 2007; Wu et al., 2016). As a large mid-troposphere heat source during summer, the TP also plays an important role in the onset and maintenance of the Asian summer monsoon (Li and Yanai, 1996; Wu and Zhang, 1998; Yihui and Chan, 2005; Wu et al., 2012), which provides almost 80 % of the precipitation in the central and eastern parts of the Himalayas during the monsoon season (June-September) (Bookhagen and Burbank, 2010; Palazzi et al., 2013; Sabin et al., 2020). In contrast, winter precipitation contributes nearly half of the annual precipitation in the Karakoram and the Hindu Kush, mostly due to the westerly disturbances (WDs) bringing moisture from the Atlantic ocean, Mediterranean and Caspian Seas (Singh et al., 1995; Vandenberghe et al., 2006; Palazzi et al., 2013; Kapnick et al., 2014; Madhura et al., 2015; Cannon et al., 2015; Hunt et al., 2018; Krishnan et al., 2019). TP and HMA are often referred as the "Asian Water Tower" and/or the "Third Pole" (e.g., Immerzeel et al., 2010; Qiu, 2008; Yao et al., 2012, 2019) because they are the largest freshwater resource stored in the cryosphere after the polar ice sheets. In this region, snowmelt ensures a permanent water flow to the major Asian river systems, such as the Yangtze, Yellow, Salween, and Mekong rivers (Sharma et al., 2019), contributing to the water supply of over 1.4 billion living downstream (Immerzeel and Bierkens, 2012; Yao et al., 2012; Rasul, 2014; Scott et al., 2019; Wester et al., 2019).

Over 1955-1996, Liu and Chen (2000) estimated an annual warming rate over the TP of 0.16 °C decade$^{-1}$ that reached 0.32 °C decade$^{-1}$ in winter, while Wang et al. (2008) observed an annual warming of 0.36 °C decade$^{-1}$ over 1960-2007. Precipitation and snow cover show contrasted trends over the TP, depending on the location and the period (Kang et al., 2010). Increasing temperature induced a reduction of the snow cover fraction in HMA, but this one has been compensated by an increase in precipitation leading to stronger snowfall rates in some regions (Viste and Sorteberg, 2015; Notarnicola, 2020). Upon its impact on snow cover, climate change in HMA and TP affects also the permafrost and the glaciers (Yang et al., 2010; Yao et al., 2007), increases the desertification (Xue et al., 2009), and affects the hydrological cycle inducing serious threats for the water resources used for agriculture, drinking water, and hydroelectricity (Qiu, 2008; Immerzeel et al., 2010; Sabin et al., 2020). HMA is also facing an increase in both the intensity and the frequency of heatwaves (Ding et al., 2018). The lack of observations, especially pronounced in the western part of HMA, limits the possibility to understand and anticipate the climate change in this area (Orsolini et al., 2019).

Current coupled ocean–atmosphere general circulation models (GCMs) have a too coarse spatial resolution (from fifty to few hundreds of kilometers) to reproduce the small-scale variability of temperature, precipitation and snowpack that is observed over complex topography areas. Nevertheless, they may be effective in providing a smooth but consistent picture of the large-


scale temporal and spatial patterns of these key variables at the regional scale. The Coupled Model Intercomparison Project (CMIP) organized by the World Climate Research Programme (WCRP), recently distributed under its sixth phase CMIP6 (Eyring et al., 2016), is a unique opportunity to conduct comprehensive analyses of the climate variability and change at both global and regional scales, based on an ensemble of climate models.

       GCM experiments show generally good skill for surface temperature, however, a systematic cold bias over TP and mountain-
ous areas has been pointed in GCM outputs since the first AMIP experiments (Mao and Robock, 1998). Su et al. (2013) showed that most of the CMIP5 models have a cold bias at the surface in the eastern TP, with a mean underestimation of $-1.1$ °C to $-2.5$ °C over December to May, and less than $-1$ °C over June to October in comparison to ground observations, while the annual climatology of precipitation is overestimated by 62 % to 183 %. Regional climate models show similar cold biases, a deficiency that is often associated with an excess of precipitation in the experiments (Lee and Suh, 2000). However, the lack
of high-elevation observation station data may also be partly responsible for the apparent cold bias of the model (Gu et al., 2012), and high-resolution experiments suggest that the real precipitation rates occurring at high elevation are likely stronger than those estimated from gridded products based on rain gauge measurements (Dimri et al., 2013). GCMs show cold biases also at 500 hPa, which may be caused by penetration of dry and cold air from the deserts of western Asia due to an overly smoothed representation of topography west of the TP (Boos and Hurley, 2013; Xu et al., 2017). Chen et al. (2017) suggested
that improvements in the parameterization of snow cover area and boundary layer processes in CMIP5 models should allow to improve the representation of the surface energy budget and to reduce the cold bias over TP. Model biases are also related to inaccurate descriptions of the elevation and the atmospheric circulation as the Asian anticyclone or summer monsoon (e.g., Salunke et al., 2019; Duan et al., 2013). More recently, Zhu and Yang (2020), compared CMIP6 and CMIP5 models over 1961–2014 to finally conclude that the cold bias and the wet bias over TP, even if reduced, still persist in the most recent
version of these models.

       Our study focuses on the climate variability over HMA as simulated with CMIP6 models. The near-surface air temperature, the snow cover extent and the total precipitation are considered to answer four questions: (1) what are the biases in HMA in this new generation of climate models for these 3 variables? (2) What are the links between the model biases in temperature, precipitation and snow cover? (3) Do the model biases impact the simulated climate trends? (4) Which climate projections
can be expected in this area over the next century? The datasets and methods used in this study are described in the next section. Section 3 presents a comparison between observations and 26 CMIP6 models over the historical period with a focus on the potential correlations between the biases of the different variables. We then show the historical trends estimated from the CMIP6 experiments and their potential dependency on model biases (Section 4). Section 5 explores future projections under different scenarios covering the twenty-first century. Sections 6 and 7 provide respectively a discussion and the conclusions.



## 2 Data and methods

### 2.1 Models

In this study, we selected 26 GCMs Table 1 in the CMIP6 database (Eyring et al., 2016) focusing on near-surface air temperature (tas), total precipitation (pr) and snow cover extent (snc) over 1979-2014. Only 10 of these models are available for future projections covering the ensemble of the four shared socioeconomic pathways SSP1-2.6, SSP2-4.5, SSP3-7.0 and SSP5-8.5, which are combining socio-economic scenarios and radiative forcing levels (O'Neill et al., 2016). Considering the model uncertainties, such a limited number of models might be sufficient to explore future climate trends (Knutti et al., 2010). The resolution of the models ranges from about 3° to 0.5° (∼300 km to 50 km) while most of them reach a 1° resolution. All models are regridded on a common 1° x 1° grid using a bilinear interpolation before the multimodel analysis.

Climatologies are computed with the first member (usually *r1i1p1f1*), whereas trend analyses are based on ensemble means for each model, restricted to a single model physics parameter set (p), initialization method (i) and forcing (f), except when a different recommendation is given by the modeling group (Table A1). The version of the model data is the most recent one available at the time of this analysis.

### 2.2 Observations

Because of complex topography, severe weather and harsh environmental conditions in HMA and TP, meteorological observations are rare in this region. Available weather stations are usually sparse and unevenly distributed (Wang and Zeng, 2012; Su et al., 2013). Gridded data, satellite observations and reanalyses are combined here to obtain a robust evaluation of model biases, even if affected by the uncertainties inherent to the observations.

#### 2.2.1 Near-surface air temperature

The CRU TS (Climatic Research Unit gridded Time Series) version 4.00 (http://doi.org/10/gbr3nj) provides a 0.5° gridded dataset of the monthly temperature (excluding Antarctica) available from 1901 until present, based on local weather stations and provided with an estimation of the data quality (Harris et al., 2020). This dataset has been widely used over HMA and TP (e.g., Gu et al., 2012; Chen et al., 2017; Krishnan et al., 2019; Wang et al., 2021; Yi et al., 2021). Correlation with local measurements, including at high elevation (Wang et al., 2013b), is high in this region. This gives confidence for model evaluation (Chen et al., 2017).

#### 2.2.2 Snow cover extent

In situ snow observations are sparse over HMA and TP, and when they are available, in situ data are often not representative for snow cover analysis at the regional scale (Gurung et al., 2017). Alternatively, remote sensing datasets provide large-scale snow information useful for spatiotemporal analyses. The satellite product available over the longest period, but at a coarse spatial resolution, is the NOAA Climate Data Record (CDR) (Robinson et al., 1993, 2012; Estilow et al., 2015), covering the





| Institute (country) | Model | Resolution (lon x lat) | Primary member | All SSPs available | Reference |
|---|---|---|---|---|---|
| BCC (China) | BCC-CSM2-MR | 1.1° x 1.1° | r1i1p1f1 | x | Wu et al. (2019) |
| | BCC-ESM1 | 2.8° x 2.8° | | | |
| CAS (China) | CAS-ESM2-0 | 1.4° x 1.4° | r4i1p1f1 | | Zhang et al. (2020) |
| NCAR (USA) | CESM2 | 1.2° x 0.9° | r1i1p1f1 | | Danabasoglu et al. (2020) |
| | CESM2-FV2 | 2.5° x 1.9° | | | |
| | CESM2-WACCM | 1.2° x 0.9° | | | |
| | CESM2-WACCM-FV2 | 2.5° x 1.9° | | | |
| CNRM-CERFACS (France) | CNRM-CM6-1 | 1.4° x 1.4° | r1i1p1f2 | x | Voldoire et al. (2019) |
| | CNRM-CM6-1-HR | 0.5° x 0.5° | | | |
| | CNRM-ESM2-1 | 1.4° x 1.4° | | | Séférian et al. (2019) |
| CCCma (Canada) | CanESM5 | 2.8° x 2.8° | r3i1p2f1 | x | Swart et al. (2019) |
| NOAA-GFDL (USA) | GFDL-CM4 | 1.2° x 1.0° | r1i1p1f1 | | Held et al. (2019) |
| NASA-GISS (USA) | GISS-E2-1-G | 2.5° x 2.0° | r1i1p1f1 | | Kelley et al. (2020) |
| | GISS-E2-1-H | | | | |
| MOHC (UK) | HadGEM3-GC31-LL | 1.9° x 1.2° | r1i1p1f3 | | Andrews et al. (2020) |
| | HadGEM3-GC31-MM | 0.8° x 0.6° | | | |
| IPSL (France) | IPSL-CM6A-LR | 2.5° x 1.3° | r1i1p1f1 | x | Boucher et al. (2020) |
| MIROC (Japan) | MIROC-ES2L | 2.8° x 2.8° | r1i1p1f2 | x | Hajima et al. (2020) |
| | MIROC6 | 1.4° x 1.4° | r1i1p1f1 | | Tatebe et al. (2019) |
| MPI-M (Germany) | MPI-ESM1-2-HR | 0.9° x 0.9° | r1i1p1f1 | | Gutjahr et al. (2019) |
| | MPI-ESM1-2-LR | 1.9° x 1.9° | | | Mauritsen et al. (2019) |
| MRI (Japan) | MRI-ESM2-0 | 1.1° x 1.1° | r1i1p1f1 | x | Yukimoto et al. (2019) |
| NCC (Norway) | NorESM2-LM | 2.5° x 1.9° | r2i1p1f1 | | Seland et al. (2020) |
| SNU (South Korea) | SAM0-UNICON | 1.2° x 0.9° | r1i1p1f1 | | Park et al. (2019) |
| AS-RCEC (Taiwan) | TaiESM1 | 1.2° x 0.9° | r1i1p1f1 | | Lee et al. (2020) |
| MOHC (UK), NIMS-KMA (South Korea) | UKESM1-0-LL | 1.9° x 1.2° | r1i1p1f2 | x | Sellar et al. (2019) |

**Table 1.** Description of the CMIP6 models used in this study with their institute, name, approximate spatial resolution (longitude x latitude), the member considered in the 1-member analyses and their reference. A cross is included in the last column when the model projections are available for the four SSPs scenarios (SSP1-2.6, SSP2-4.5, SSP3-7.0 and SSP5-8.5).





Northern Hemisphere (NH) from October 4, 1966 to present (referred to as NOAA CDR in this article). Data prior to June 1999 is based on weekly satellite-derived maps of snow cover extent (SCE), whereas posterior data has been replaced by daily SCE estimated from the Interactive Multisensor Snow and Ice Mapping System (IMS). The weekly SCE maps are digitized to a 88x88 grid following a 190 km polar stereographic projection and contain binary snow cover information. The retrieval of snow cover information for this product is not interfered by clouds due to the weekly aggregation prior to June 1999 and the

inclusion of passive microwave data posterior. The NOAA CDR has been widely used in climate-snow studies over the NH (e.g., Brown and Robinson, 2011; Hernández-Henríquez et al., 2015; Hori et al., 2017; Santolaria-Otín and Zolina, 2020) and more specifically over HMA (e.g., Xu et al., 2016). This dataset is adapted for continental-scale studies, but shows limitations over mountainous regions (Déry and Brown, 2007), even if the inclusion of Meteosat-5 data in 2001 significantly improved its quality over the Asian continent (Helfrich et al., 2007). Trend analyses based on NOAA CDR data must be taken with caution

because of potential temporal heterogeneities related to changes of experimental protocols (Mudryk et al., 2020). To obtain monthly fractional values, we simply average the weekly binaries values included in each corresponding month.

    The AVHRR GAC snow cover extent time series version 1 derived in the frame of the ESA CCI+ Snow project is the most recent long-term global snow cover product available (Naegeli et al., 2021). It covers the period 1982-2020 at a daily temporal and 0.05° spatial resolution. The product is based on the Fundamental Climate Data Record (FCDR) consisting of

daily composites of AVHRR GAC data (https://doi.org/10.5676/DWD/ESA_Cloud_cci/AVHRR-PM/V003) produced in the ESA Cloud CCI project (Stengel et al., 2020). The data were pre-processed with an improved geocoding and an inter-channel and inter-sensor calibration using PyGAC (Devasthale et al., 2017). Alongside the daily reflectance and brightness temperature information, an excellent cloud mask including pixel-based uncertainty information is provided (Stengel et al., 2017, 2020). Snow cover extent was retrieved using SCAmod (Metsämäki et al., 2015), while water bodies, permanent ice bodies and

missing values are flagged. To reduce the effect of cloud coverage, a temporal filter of ±3 days of each individual snow cover observation was applied after Foppa and Seiz (2012). The AVHRR GAC FCDR snow cover product comprises only one longer data gap of 92 days between November 1994 and January 1995 resulting in a 99 % data coverage over the entire study period of 38 years. For the computation of the average annual cycle over the study period, the permanent ice bodies were assumed to be 100 % snow covered, whereas water bodies, remaining clouds or other missing values were not taken into account. Due

to the slightly shorter time period covered by this snow product compared to the period investigated in this study, it was not considered for trend analysis.

### 2.2.3   Precipitation

In this study, we use the daily APHRODITE (Asian Precipitation-Highly-Resolved Observational Data Integration Towards Evaluation of Water Resources) product (Yatagai et al., 2012) version V1101 (1951 - 2007) and its extended version V1101EX_R1

(2007 - 2015) over the domain Monsoon Asia (MA) at a 0.5° resolution. APHRODITE includes a large number of local observations and includes a correction in the interpolation process for complex topography areas. The seasonal precipitation is correctly represented in APHRODITE (e.g., Palazzi et al., 2013; Kapnick et al., 2014). However, most of the stations are located in the eastern and southern parts of the TP, and do not cover the high elevation areas. For comparison, we used the



Global Precipitation Climatology Project (GPCP) Climate Data Record (CDR), Version 2.3 (Monthly) product at 2.5° (Adler
et al., 2016, 2018). This product combines satellite products with rain gauge stations available from 1979 to present. However,
the scarcity of high-elevation in situ stations, the interference of wind with the sensors, and the problems of satellite-based
meteorological radars in identifying snow crystals lead to large uncertainties in observational snowfall datasets (Palazzi et al.,
2013; Sun et al., 2018). Total precipitation is therefore generally underestimated, especially over snow rich areas (Sanjay et al.,
2017).

### 2.2.4 Topography

Global Multi-resolution Terrain Elevation Data 2010 (GMTED2010) (Danielson and Gesch, 2011) (available at https://www.
temis.nl/data/gmted2010/index.php) provides elevations and its standard deviation at multiple resolutions and is realistic over
HMA (Grohmann, 2016). In this study, we use the 1° x 1° resolution as a reference grid.

## 2.3 Reanalyses

Reanalysis data, based on assimilation of meteorological observations, provide an estimate of the climate variability at the
global and regional scales consistent with the observed variability. An advantage over most observations is that reanalysis data
do account for total precipitation, providing separately the rainfall and snowfall rates (Palazzi et al., 2013). However, climate
trends estimated from reanalysis data are affected by the continuous changes in the observing systems that can introduce
spurious variability and trends (Bengtsson, 2004). Global atmospheric reanalyses show poor quality over HMA and TP also
because of their coarse resolution and the limited number of local observations available for the assimilation process that is not
adapted for such complex topography areas (You et al., 2010; Norris et al., 2015, 2017).

### 2.3.1 ERA-Interim

ERA-Interim is a global atmospheric reanalysis dataset produced by the European Centre for Medium-Range Weather Forecasts
(ECMWF), covering the period from 1979 to 2019 at approximately 80 km on 60 vertical levels (Dee et al., 2011). ERA-
Interim shows best overall performances on air temperatures compared to other reanalyses over TP (Wang and Zeng, 2012)
and high correlations (0.97 to 0.99) with respect to ground meteorological stations during 1979–2010 (Gao et al., 2014).
Estimates of precipitation associated with the reanalysis are produced by the forecast model, based on the assimilation of
temperature and humidity observations (Palazzi et al., 2013). Snow depth is assimilated through station observations (Orsolini
et al., 2019) and gridded snow cover from IMS is also assimilated since 2004 (Drusch et al., 2004). As described in the ECMWF
documentation[1], snow cover fraction is a diagnostic variable computed directly using snow water equivalent (ie parameter SD
in m of water equivalent) as $SCF = \min(1, RW \times SD/15)$, where RW is the density of water equal to 1000.

---

[1]https://confluence.ecmwf.int/display/CKB/ERA-Interim%3A+documentation#ERAInterim:documentation-Computationofnear-surfacehumidityandsnowcover



### 2.3.2 ERA5

ERA5 is the most recent global atmospheric reanalysis produced by the ECMWF and replaces ERA-Interim (Hersbach et al., 2020). The improvements including the spatial and temporal resolution (hourly estimates at 31 km distributed on 137 levels),

allowed for example an improved representation of the troposphere and better global balance of precipitation and evaporation. As in ERA-Interim, snow cover fraction is a diagnostic variable that can be computed from snow water equivalent (ie parameter SD in m of water equivalent) and RSN as $SCF = \min(1, (RW \times SD/RSN)/0.1)$, where RW is the density of water equal to 1000. Unlike ERA-Interim, IMS data are not used above 1500 m, i.e. in high altitude regions, which includes the TP (ECMWF, 2020).

## 2.4  Study area

In this study we consider HMA as a box covering 60° E - 110° E and 20° N - 45° N (Fig. 1 a and b) focusing on mountain areas, including the TP, with an elevation higher than 2500 m. As in previous studies considering different climatic areas (e.g., Palazzi et al., 2013; Kapnick et al., 2014; Sanjay et al., 2017), three subdomains are considered: Hindu-Kush Karakoram (HK; 70° E - 81° E; 31° N - 40° N), Himalayas (HM; 79° E - 98° E; 26° N - 31° N) and Tibetan Plateau (TP; 81° E - 104°E;

31° N - 39° N) using grid cells within each sub-region above 2500 m. HK is largely influenced by WDs, whereas most of the precipitation over HM is related to the Asian summer monsoon. A cold and dry continental climate is found in TP (Bookhagen and Burbank, 2010; Palazzi et al., 2013; Sabin et al., 2020).

## 2.5  Numerical methods and computations

Trend computations are based on linear least-squares regression. We consider a 95 % level of significance, corresponding to a

p-value equal to 0.05, computed with a two-sided Wald test for which the null hypothesis corresponds to a slope equal to zero. The linear relationship between two datasets is estimated with a Pearson correlation. For spatial correlations, the values are flattened to one dimension before applying the computation.

The cosine latitude is taken into account as a weight in spatial averages and the exact number of days in each month, depending on the calendar type, is considered in temporal averages. Our analyses cover the historical period 1979-2014 and

projections over 2015-2100, focusing on two seasons: the summer extending from June to September (JJAS), a period when the monsoon is active (Palazzi et al., 2013; P Sabin et al., 2013), and the winter defined as the months covering December to April (DJFMA), a period affected by WDs precipitation especially pronounced over the Hindu-Kush and Karakoram areas (Palazzi et al., 2013; Kapnick et al., 2014; Cannon et al., 2015; Hunt et al., 2018; Krishnan et al., 2019). Annual means are also considered when seasonal analysis does not show additional information.



For model evaluation we use two different metrics based on spatial climatologies: the root mean square error (RMSE; Eq. 1) and the mean bias (Eq. 2), that we slightly modified to take into account the spatial weight ($w$; Eq. 3) of each grid cell.

$$\text{RMSE} = \sqrt{\frac{1}{\sum_{i=1}^{n} w_i} \sum_{i=1}^{n} w_i \left(M_i - O_i\right)^2} \tag{1}$$

$$\text{Mean Bias} = \frac{1}{\sum_{i=1}^{n} w_i} \sum_{i=1}^{n} w_i \left(M_i - O_i\right) \tag{2}$$

$$w = \cos\lambda \tag{3}$$

where $\lambda$ is the latitude, $M_i$ represents model simulations and $O_i$ the observed data.

To characterize the multimodel ensemble, mean or median are usually considered in addition to their 5th and 95th percentiles (e.g. mean/median [5th 95th]). Multimodel mean is used in bias analysis and projections are based on the multimodel median.

## 3 Historical bias analysis

Model biases are computed with the observation datasets CRU, APHRODITE and NOAA CDR used as references for near-surface air temperature, total precipitation and snow cover extent respectively over the period 1979-2014. Whenever possible, we show a comparison with other datasets to get more confidence in the model bias quantification.

### 3.1 Climatologies

The annual climatology computed over 1979-2014 is shown in Fig. 1 for the CRU, NOAA CDR and APHRODITE observations (c, e, g) and the multimodel mean based on 26 CMIP6 models (d, f, h). Over HMA, temperature ranges from −8 °C in high elevation areas to 13 °C at lower elevation in observations with an average of −0.2 °C (Fig. 1c). HMA temperature reaches −15 °C to 9 °C in winter and 2 °C to 19 °C in summer (not shown). The multimodel mean shows colder temperature than observations, with values ranging from −11 °C to 3 °C and a mean value over HMA about −2.1 °C (Fig. 1d). Even with a general cold bias, the spatial pattern of temperature in the model is consistent with the observations, with a spatial correlation of 0.86.

Snow cover extent is heterogenous over HMA (Fig. 1e), with high values over HK reaching more than 70 % that are explained by strong winter snowfalls related to WDs (Cannon et al., 2015; Bao and You, 2019). Snow cover extent is much smaller over most of the TP region with annual values not exceeding 20 %. High values, around 50 %, are also found over the Tien Shan and the southeast Himalaya. In the multimodel mean (Fig. 1f), snow cover is overestimated over most of the TP, and slightly underestimated over the HK region in comparison with the observations.





Strong precipitation rates, reaching an annual mean of more than 6 mm d$^{-1}$(exceeding 2000 mm per year), are observed
in the eastern part of HM, mostly due to the Asian summer monsoon, with a decreasing influence from the southeast to
the northwest Himalayan chain (Fig. 1g). In contrast, the HK region receives moisture from both Asian summer monsoon
and WDs (Fig. 2j). Moisture-laden westerly winds are intercepted by high mountain ranges in northern Pakistan, leading to
moisture condensation and precipitation at high elevation (Palazzi et al., 2013), partly explaining the high values of snow

cover in this area Fig. 1e). Due to the orographic barrier, the TP located more on the East is much drier, with annual mean
precipitation generally lower than 1 mm d$^{-1}$. The multimodel mean (Fig. 1h) shows globally higher values of precipitation
over HMA in comparison with the observations. Precipitation tends to spread more over the TP in the model compared to the
observations, which might be due partly due to the smoothing of the topography in the models. However, precipitation rates
are also generally underestimated in observational datasets because of snowfall undercatch issues, which could lend credence

in the stronger precipitation rates modeled at high elevation.

## 3.2  Temperature, snow cover and precipitation annual cycle

The seasonal cycles are shown in Fig. 2 for the models and different observational datasets and reanalyses over HMA and the
three sub-domains for temperature, snow cover and precipitation. The model biases with respect to observations are stronger in
winter than in summer for temperature and snow cover, a feature already noticed in CMIP5 and CMIP6 (e.g., Su et al., 2013;

Zhu and Yang, 2020). Indeed, the multimodel mean temperature is around 2 to 3 °C below the CRU observations in winter over
HMA, while models and observations are much closer in summer (Fig. 2a). These differences are more pronounced in the HK
region (Fig. 2b) with differences noticed both in winter (4 to 5 °C) and summer (∼2 °C). For temperature, ERA-Interim is very
close to the CRU while ERA5 is closer to the multimodel mean, except for the HK region where both reanalyses underestimate
temperature in comparison with the CRU.

In comparison with the NOAA CDR satellite observation, the multimodel mean snow cover over HMA is overestimated by
20 % in winter and is closer to the observations in summer when snow cover is lowest (Fig. 2e). The model spread is larger
for snow cover than for temperature and precipitation, with values varying from 20 % to 90 % in winter and 0 to 40 % in
summer. This large spread highlights the difficulty to simulate snow cover in complex topography areas and also the large
internal variability of snow cover. ERA-Interim is again very close to the observations for snow cover, likely because of the

assimilation of IMS data in this reanalysis, a satellite product also used in the production of the NOAA CDR dataset (Drusch
et al., 2004; Robinson et al., 2012). The more recent ECMWF reanalysis ERA5 shows an overestimation of snow cover that
is comparable to the CMIP6 model ones. This behavior has already been described in Orsolini et al. (2019), explaining this
difference by the fact that ERA5 does not assimilate IMS data beyond 1500 m asl, while Hersbach et al. (2020) suggests that
the single-layer snow scheme does not allow enough melting in mountainous regions. The differences found over the three

subdomains are similar to those highlighted over the whole HMA (Fig. 2f-h). Nevertheless, we note a precocious spring melt
in the multimodel mean and ERA-Interim compared to the NOAA CDR observations and ERA5 in HK region (Fig. 2f). The
ESA CCI product shows lower snow cover values compared to NOAA CDR and ERA-Interim, with values around 30 %
during the winter in HMA, suggesting that model biases may be even larger. The large difference in spatial resolution with

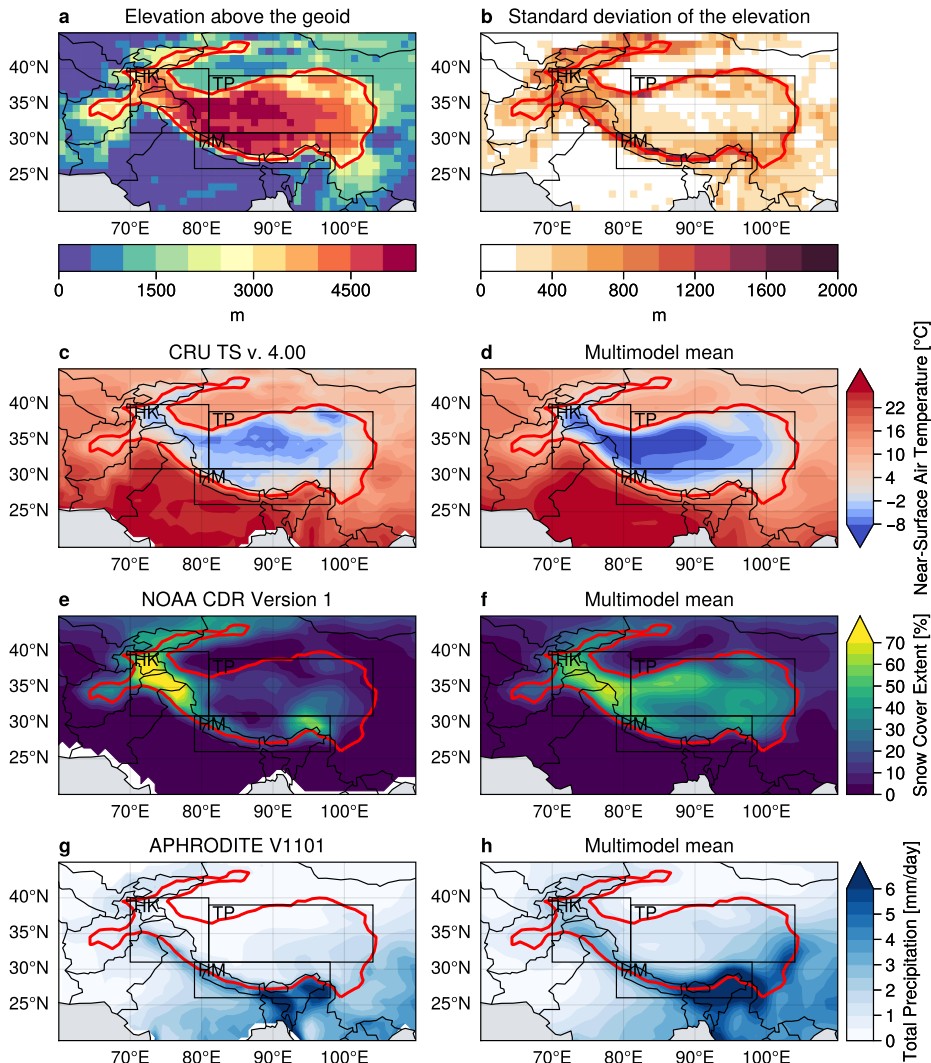

**Figure 1.** Surface elevation (a) and its standard deviation (b) estimated from GMTED2010 at 1°. Annual climatologies computed over 1979-2014 for temperature (c, d), snow cover(e, f) and precipitation (g, h); the left panels correspond to the observations CRU (c), NOAA CDR (e), and APHRODITE (g) while the right panels (d, f, h) correspond to the multimodel mean (using the first realization of each ensemble model). The red contour highlights the HMA domain limited to areas higher than 2500 m asl and the black boxes define the sub-domains Hindu-Kush Karakoram (HK), Himalayas (HM), and Tibetan Plateau (TP), that are also limited to areas higher than 2500 m asl (red contour) in this study.

the latter product may also play a role in this discrepancy, as valleys and other aspects of fine scale topography are not being
well resolved in the other products and thus lack a good representation of the spatial heterogeneity of snow cover compared to the ESA CCI product. This suggest a general snow cover overestimation in both model and reanalysis data based on coarse

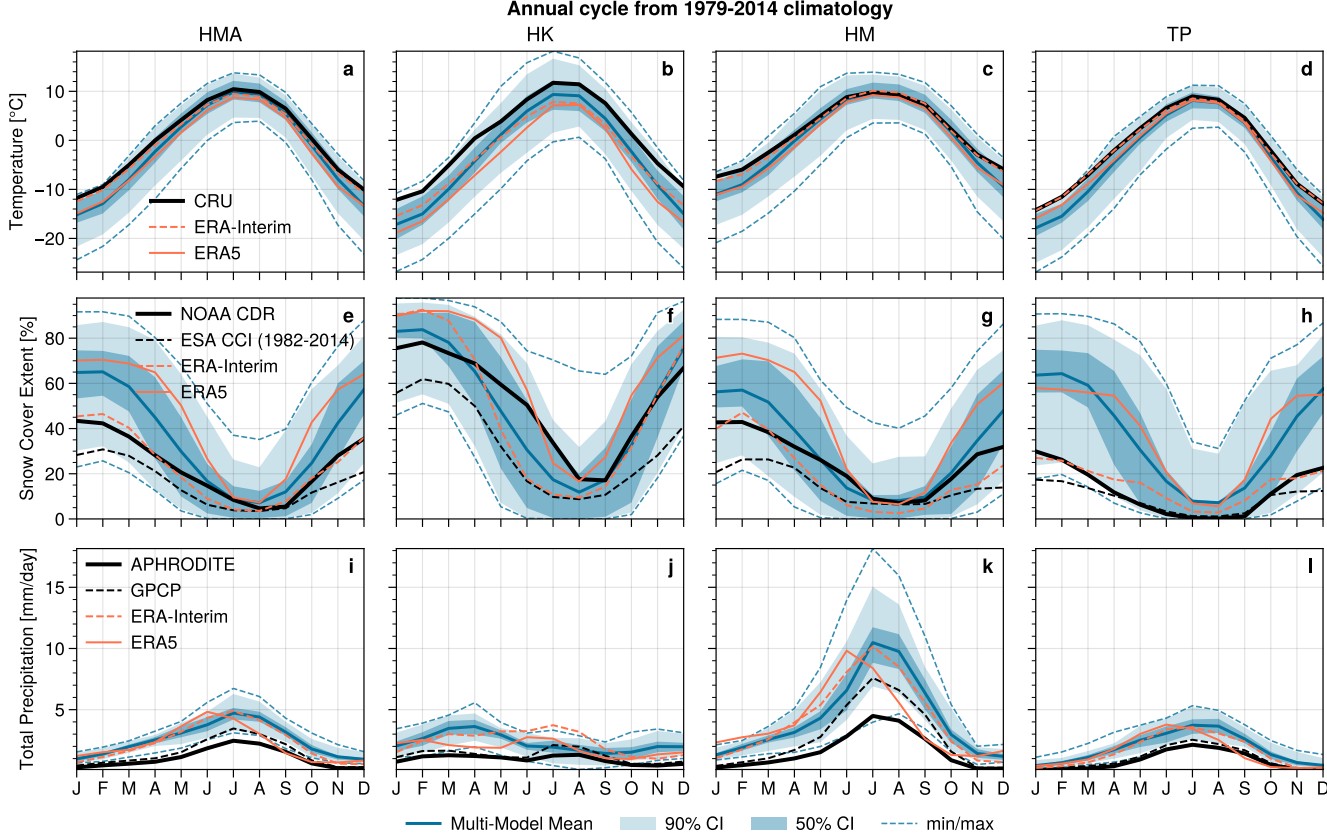

**Figure 2.** 1979-2014 climatology of the annual cycle of temperature (a-d), snow cover(e-h) and precipitation (i-l) averaged over HMA (a, e, i) HK (b, f, j) HM (c, g, k) and TP (d, h, l), excluding the surface area located below 2500 m asl (red contours in Fig. 1). The multimodel mean (dark blue line) is shown with the 50 % confidence interval (CI, dark blue shading), the 90 % CI (light blue shading) and the minimum and maximum (dashed blue lines) of the ensemble. The black curves correspond to the observational datasets: CRU, NOAA CDR and APHRODITE respectively for temperature, snow cover and precipitation. The reanalysis ERA-Interim and ERA5 are shown respectively with the dashed and solid orange curves. GPCP and ESA CCI dataset are also shown for snow cover and precipitation respectively (dashed black line). The ESA CCI covers only the 1982-2014 period.

resolution, that is especially pronounced over high-mountain areas (Fig. 2f, g). Nevertheless, the positive bias of snow cover fraction simulated by the models over HMA is mainly related to the overestimation of the snow cover over TP, where the snow cover varies around 20 % for the observations, while values above 60 % are found in the multimodel mean, despite a wide

dispersion among the models (from 20 to 90 %).

The strongest precipitation rates occur during the Asian summer monsoon over the HM region (> 2500 m) with precipitation rates reaching a monthly mean of 10 mm d$^{-1}$ (~300 mm per month) on average in the multimodel mean, while precipitation rates reach lower values than 2 mm d$^{-1}$ during the winter (Fig. 2k). Other regions exhibit smaller precipitation amounts below



5 mm d$^{-1}$ most of the year. In the HK region, larger precipitation rates are found in late winter and spring (Fig. 2j) mostly

due to WDs as explained in Sect. 1. While the model spread generally encompasses the observations for temperature and snow cover, APHRODITE precipitation data is most of the time below the minimum of the model values. This difference might be explained by snow undercatch issues typically obtained with rain gauge measurements (e.g., Jimeno-Sáez et al., 2020), whereas models are expected to provide both solid and liquid precipitation. In addition, rain gauge measurements are generally too sparse to estimate the heterogeneous distribution of precipitation over complex topography areas. Over HK, a large part

of the precipitation falls as snow in winter, a period when strong differences between satellite/rain gauge products (black curves) and models/reanalyses are also appearing during February to May, whereas the precipitation is closer between models and observations during the summer (Fig. 2j). Nevertheless, GPCP data (black dash line) is slightly closer to the models, especially over the HM domain. ERA5 has an early precipitation peak in June, while it is found in July for the other products. Precipitation datasets should be considered carefully knowing that there is no better product than the other ones in this region,

and the effective values of precipitation rates are highly uncertain in this area (Palazzi et al., 2013).

### 3.3 Near-surface air temperature bias

The pattern of the temperature bias widely differs from one model to another Fig. 3). However, most of the models show a cold bias, which is reflected by the multimodel mean reaching an average bias of $-1.9$ [$-8.2$ to $2.9$] °C. The cold bias is generally more pronounced at high elevation (Fig. 1a). The largest biases are found for the CNRM and IPSL models, with biases reaching

almost $-10$ °C on average and exceeding locally $-12$ °C, especially over the western part of the TP and in the Karakoram area (HK region). The other models show slight positive or negative biases around $\pm 3$ °C.

   The bias of snow cover (Fig. B1) and precipitation (Fig. B2) show contrasted patterns. The annual multimodel mean of snow cover is overestimated by 12 [$-13$ to $43$] % (or 52 [$-53$ to $183$] % relatively) over HMA compared to NOAA CDR and can reach locally an absolute difference of 40 %, while a minority of models show a slight underestimation of snow

cover (e.g. MPI-ESM1-2-HR, MPI-ESM1-2-LR, NorESM2-LM). Correlation between temperature and snow cover bias is not obvious for all the models, since some of them have a warm (cold) bias superimposed to an overestimation (underestimation) of snow cover (e.g. MIROCC-ES2L, HadGEM3-GC31-LL, HadGEM3-GC31-MM). All models show higher precipitation rates in comparison with APHRODITE (Fig. B2), as seen in the annual cycles (Fig. 2). Indeed, the multimodel annual mean bias of precipitation over HMA is 1.5 [$0.3$ to $2.9$] mm d$^{-1}$ (or 143 [$31$ to $281$] % relative to APHRODITE). The bias pattern is

consistent with the climatological pattern, with stronger absolute precipitation biases in the southeast Himalaya, where high precipitation rates are observed.

### 3.4 Spatial bias correlation

To investigate the potential links between the biases of the different variables, the correlation patterns between the biases of temperature, snow cover, precipitation and surface elevation are shown in (Fig. 4). For most of the models, a significant

negative correlation is found between the biases of temperature and snow cover, highlighting the influence of these 2 variables on each other. However, it is not possible to deduce whether it is the snow cover bias that induces the temperature bias or the





opposite. The strongest correlations between temperature and snow cover are found for the IPSL-CM6A-LR and the MIROC-ES2L models, suggesting that these biases are exacerbated by feedbacks between these two variables, while lower correlations are found with precipitation biases. On the contrary, some models (e.g. HadGEM3-GC31-MM) show surprisingly a positive
correlation between temperature and snow cover, suggesting that other processes can play a role in the development of biases (e.g. aerosol deposition on snow, cloud cover, tropospheric biases, etc.).

The correlations between the biases of temperature and precipitation are generally weaker, but with negative and significant values between $-0.12$ and $-0.37$ (except for CanESM5 which has a positive correlation of 0.16). This seems counter-intuitive as we generally expect precipitation rates to increase with temperature unless dynamical changes of the atmosphere could
induce an opposite signal at the regional scale. However, the positive precipitation model biases are likely due to the underestimation of solid precipitation in the APHRODITE observation, which would suggest an unrealistic excess of precipitation in the models. Therefore, these negative correlations are potentially not reliable and have to be considered carefully. The comparison with GPCP shows correlations between the biases of temperature and precipitation that reach positive values for a higher number of models Fig. C1. These correlations would likely be more positive if we used an observational reference dataset that was
not affected by snow undercatch issues. Nevertheless, the BCC-ESM1 and CAS-ESM2-0 models show a strong correlation between snow cover and precipitation biases (0.48 and 0.41 respectively). This link is particularly striking for CAS-ESM2-0, for which the biases of snow cover and precipitation show similar patterns over the TP and HM (Fig. B1 and Fig. B2), suggesting that snow cover biases in that case are partly due to an excess of precipitation.

The temperature and snow cover biases correlations with the surface elevation show a more uniform behavior among the
models. In general, an anti-correlation between temperature bias and elevation is found, whereas snow cover correlates positively with the elevation. The higher the elevation, the greater the biases for temperature and snow cover, suggesting that the models have difficulty representing physical processes at high elevation. The link between precipitation bias and elevation is less pronounced with fewer significant correlations (e.g., BCC-ESM1, CNRM-CM6-1-HR, HadGEM3-GC31-MM), which can either be positive or negative.

These spatial correlations are of course region and season dependent. For example, we observe stronger correlations between precipitation and snow cover biases in winter over TP, while the latter is stronger in summer over HK for most of models (Fig. C2 and C3). This may be related to an excess of moisture supply over TP in winter, due to the lack of orographic barrier effect because of the coarse resolution of the models, resulting in too much snow accumulation which is more likely to persist due to winter cold temperatures over the TP. Concerning the excess of summer precipitation in HK, this may be due to an
overextension of precipitation towards the west of the Himalayan mountain range during the monsoon period. However, this last correlation, supporting the idea that the excess snow cover may be due to excess precipitation for some models, does not necessarily explain the cold bias at the surface. For example, the HadGEM3 models have strong significant correlations between snow cover and precipitation biases over the TP (0.63 and 0.66 in annual), but do not show a significant surface cold bias (-0.12 and -0.18 °C; Fig. C3). The relationship of the biases with altitude is not always verified either, especially for
models showing warm biases such as the CESM2 family of models over the TP in summer.


## 3.5  Metrics

Spatial RMSE and mean biases are computed over HMA for the 26 models (Fig. 5). CESM2, CESM2-WACCM and MPI-ESM1-2-HR show the lowest temperature RMSE (∼2.5 °C), with a mean bias smaller than 1 °C, while worse performing models are CNRM-CM6-1, IPSL-CM6A-LR and CNRM-CM6-1-HR with RMSE exceeding 7 °C. The best models for tem-
perature are not necessarily the best ones for snow cover and precipitation (e.g. HadGEM3-GC31-LL, HadGEM3-GC31-MM, UKESM1-0-LL). RMSE for snow cover ranges from about 10 % to 45 % and most of the models show a positive snow cover bias over HMA. RMSE for precipitation ranges from over 1 mm d$^{-1}$ to 3.5 mm d$^{-1}$ while mean biases are all positive ranging from about 0.5 mm d$^{-1}$ to slightly over 2 mm d$^{-1}$ (Fig. 5c), as we already discussed in Sect. 3.3.

On the right panels of Fig. 5 (b, d, f), the RMSE and mean bias are ranked by model resolution. Finer resolution models do
not show better skill for temperature, snow cover and precipitation, suggesting that GCM resolution is not the more important criterion for climate modeling over this region. This general assumption is not the case for all model families. For example, the MPI-ESM1-2-LR (1.9° x 1.9°) and MPI-ESM1-2-HR (0.9° x 0.9°) does show slight improvements for all variables with increasing resolution. However, for the CNRM-CM6-1 (1.4° x 1.4°) and its high-resolution version CNRM-CM6-1-HR (0.5° x 0.5°), the increase in resolution leads to a degradation for temperature and snow cover, while there is a slight improve-
ment for precipitation.

None of the models stand out with respect to the others focusing on the RMSE and bias metrics. This finding suggests that there is no reason to exclude some models for climate analysis purposes in this area in particular when looking at future projections. Nevertheless, to go deeper in this question, the potential relationship between biases and trends is investigated in Sect. 4.2.

# 4  Historical trends analysis

Disentangling the trends related to internal variability from the forced signals related to anthropogenic forcing is challenging. At mid latitudes, internal variability can contribute until 50 % to climate trends computed over 50 years (Deser et al., 2020). However, this contribution decreases when considering areas closer to the tropics and when integrating climate signals over large domains (Hawkins et al., 2016). The climate trends in HMA are explored in this section over the period 1979-2014,
by comparing observational datasets and multimodel mean computed with a single member for each model to give the same weighting to each model (Fig. 6). This comparison should be considered carefully, since observational datasets reflect the superposition of both the internal variability and the forced signals whereas the internal variability is partly filtered out when averaging the model outputs. However, the 35-year period considered here is supposed to be long enough to exclude a large part of the internal variability. The spatial comparison between models and observations is used in Sect. 4.1 to investigate
forced signals and model deficiencies at the regional scale.

The modulation of the trends by internal variability is explored in Sect. 4.2 where trends are spatially integrated separately for each ensemble member to investigate the contribution of internal variability and to investigate the potential impact of model biases on simulated trends.



## 4.1 Trends

Figure 6 shows a general positive trend for temperature in observations and models during both seasons. In winter observations, stronger temperature trends are found over the TB with values ranging from 0.3 to 0.6 °C decade$^{-1}$ over 1979-2014, while weaker warming occurred over the Indo-Gangetic plain downward of HM, with values not exceeding 0.3 °C decade$^{-1}$ (Fig. 6a). The multimodel mean shows slightly lower values of temperature trends in winter than the CRU observations, except for the northern HMA. Summer temperature trends show a similar pattern as the winter ones, but with the highest values over the

western part of TP, reaching 0.5 °C decade$^{-1}$ in the model, while the CRU observations show a northward shift of the positive trends close to the Tarim desert. The temperature change found in CMIP6 models and observations is consistent with previous estimations (e.g., Wang et al., 2008; Liu and Chen, 2000) and has also been highlighted in ERA-Interim and ERA reanalyses with slightly different patterns (Fig. D1).

There are more discrepancies between models and observations for snow cover (Fig. 6e-h). The multimodel mean shows a

slight significant decreasing trend of snow cover in both winter and summer, reaching −1 to −2 % decade$^{-1}$ over TP (Fig. 6f, h), while observations show more pronounced trends with a spatially heterogeneous pattern in winter (Fig. 6e) and a significant decline in snow cover in summer over the whole Himalayas, extending to HK and the Tien Shan, with values exceeding −5 % decade$^{-1}$ (Fig. 6g). Meanwhile, the observation show slightly positive trends for the TP. However, the snow cover trends observed in the NOAA CDR dataset should be taken with caution, due to the poor resolution (∼200 km) which is problematic

for mountain areas (Bormann et al., 2018).

Observed trends are generally less significant for precipitation than for temperature. The large interannual precipitation variability limits the possibility to detect long-term trends for this variable. This explains also the discrepancy between the observation and the multimodel mean for precipitation (Fig. 6i-l), with signals that show a larger amplitude in the observation than in the multimodel mean where the internal variability has been filtered out by averaging several model outputs. In the

observations, the main precipitation signal is a significant increase in HK during both seasons, which extends southward in the Indian plain during the summer, ranging from 0.1 to 0.5 mm d$^{-1}$ decade$^{-1}$ (Fig. 6i, k). This pattern is not found in the multimodel mean, whereas an increase in precipitation is simulated during the summer, but shifted eastward over HM in comparison with the observation, with values between 0.1 to 0.3 mm d$^{-1}$ decade$^{-1}$ (Fig. 6j, l). This summer signal is likely related to monsoon changes, with patterns that differ however between models and observations. During both seasons, the

multimodel mean suggests a general and significant increase in precipitation over TP, with values around 0.1 mm d$^{-1}$ decade$^{-1}$. This signal is consistent with the observation during the summer, albeit less pronounced in the model with respect to the observation, whereas there is no clear change of precipitation during the winter in the observations. Nevertheless, caution is required when considering observations of precipitation that are generally uncertain because of undercatch issues for solid precipitation (see Sect. 2.2.3). The discrepancy between different observational datasets and reanalyses trends illustrated in

Fig. D3 confirms the strong uncertainty typically found in precipitation datasets. Nevertheless, a summer signal is clearly visible in both models and observations, potentially related to monsoon changes, with an increase in precipitation rates over





wide areas of the Western part of HMA and the Indian subcontinent that is modulated by drying patterns located more on the East, that spatially diverge among the different products.

## 4.2 Trends versus bias

Figure 7 shows the ensemble trends versus the ensemble biases for each model. Most model spreads (vertical bars) generally overlap the range of the observed trends (black shading). This suggests that the forced plus the internal variability estimated with the model ensemble is compatible with the observed variability. A few model spreads stand outside of this range and should be considered carefully. However, the observed trends can also be affected by artifacts as discussed in Sect. 4.1 and are not fully reliable either. In addition, some models include only a limited number of members, sometimes a unique one

(Table A1), so the ensemble for these models does not cover the full range of plausible evolutions. Some highly biased models fall within the range of observed trends, whereas other ones show a combination of small biases and unrealistic trends. This finding suggests that model bias is not a robust criterion to discard models in trend analysis.

Nevertheless, even with a large spread, a dependency of the simulated past trends to the model biases is found for some variables and seasons, and in particular for winter temperature, summer precipitation and snow cover during both the winter

and the summer. The low values of snow cover in summer over HMA might explain the limited melting for low biased models, whereas the models with large amounts of snow cover persisting in summer show a decreasing trend of snow cover with the warming (Fig. 7e). This is likely to result in an overestimation of summer melting rates. The models overestimating the snow cover during the winter are also showing small decreasing trends of snow cover (Fig. 7b). This is likely related to attenuated warming trends found in cold biased models during the winter (Fig. 7a), suggesting a damping of the warming in cold models.

During the summer, there is no clear relationship between bias and trends of temperature (Fig. 7d), but the models simulating an excess of precipitation rates also show enhanced precipitation increases (Fig. 7f). This might be explained by the non-linear relationship between temperature and atmospheric moisture: at fixed warming rates, the models showing a more active hydrological cycle also show a stronger precipitation increase.

Even with such a relationship between biases and trends, the number of observations is too small and their uncertainties too

large to allow a robust selection of the models that could be used in climate analysis. Nevertheless, with more observations available, these results could help to select a subset of models to reduce the spread in future projections for example. Nevertheless, the relationship between models and biases is less clear when considering only the 10 models for which the full set of projections is available (orange points). We assume therefore that future trends shown in the next section are not depending on the model biases.

It should be noted that one model (CanESM5) used for projections stands out for temperature in both seasons, with values reaching about 1.3 °C decade$^{-1}$ in summer, while for the other models, summer trends range from about 0.2 °C decade$^{-1}$ to 0.5 °C decade$^{-1}$ (Fig. 7d). However, this does not justify discarding this model because recent trends are not unrealistic for other variables and seasons. Nevertheless, one can expect that the upper limit of the temperature projection range will be overestimated because of that model. Because this behavior could affect the multimodel mean, we use the median of the

multimodel ensemble instead.





## 5  Projections

In this section, we use the 10 models for which the 4 SSPs scenarios, based on different levels of anthropogenic emissions, SSP1-2.6, SSP2-4.5, SSP3-7.0, and SSP5-8.5 (O'Neill et al., 2016), are available (Table 1).

### 5.1  Projected changes over HMA

The 10 CMIP6 models project a warming over HMA at the end of this century (2081-2100 with respect to 1995-2014 average) that ranges from 1.9 [1.2 to 2.7] °C for SSP1-2.6 to 6.5 [4.9 to 9.0] °C for SSP5-8.5 (Table 2). This warming is expected to continue beyond 2100 under the SSP5-8.5 scenario (Fig. 8), while a stabilization of temperature is simulated under the scenario SSP1-2.6 between 2060 to 2080, followed by a slight cooling. This warming is associated with a snow cover extent decrease ranging from −4.4 [−10.0 to −0.1] % to −14.5 [−27.4 to −6.0] % (Table 2). This absolute change corresponds to a

relative loss of −9.4 [−16.4 to −5.0] % to −32.2 [−49.1 to −25.0] % with respect to the 1995-2014 average (Fig. 8b). These changes are concomitant with a precipitation increase from 0.2 [0.0 to 0.5] mm d$^{-1}$ to 0.6 [0.2 to 1.2] mm d$^{-1}$, corresponding respectively to a relative increase from 8.5 [4.8 to 18.2] % to 24.9 [14.4 to 48.1] % with respect to the 1995-2014 average. As for temperature, snow cover and precipitation are expected to stabilize under the SSP1-2.6 scenario while an acceleration of snow cover decrease and precipitation increase is simulated under SSP5-8.5. SSP2-4.5 and SSP3-7.0 show intermediate

pathways between SSP1-2.6 and SSP5-8.5. NOAA CDR show way greater inter-annual variability compared to the models both in the historical period and the projections, thus it seems difficult to model the natural variability of snow cover.

The future warming over HMA is more pronounced in winter than in summer (Fig. 9a-d). These seasonal contrasts are more pronounced in strong CO$_2$ emissions scenarios, with almost no differences for SSP1-2.6 (∼0.1 °C) to about 1 °C under SSP5-8.5 (5.8 °C in summer to 6.8 °C in winter; Table 2). Enhanced winter warming is associated with a strong decrease in

snow cover by −15 % (Fig. 9e-f) located over HK and northward (east side of Tien-Shan), HM and southeastern TP. This last reduction of snow cover, must be amplified by an early spring melt. The snow cover decrease is smaller in summer and is mainly located in the Western part of TP. Snow cover extent is drastically reduced in summer from about −30 % to more than −80 % locally (Table E1), and only in the areas where snow cover persists during the summer. Precipitation is projected to increase both in summer and winter. The precipitation increase in winter in HK suggests an intensification of WDs, while the

summer increase found over HM and TP corresponds to an increase in monsoon-related precipitation (Fig. 9i-l). The relative increase in precipitations is slightly higher in summer than in winter for the whole HMA domain, ranging from 6.4 [0.7 to 13.5] % to 22.8 [9.8 to 45.8] % in winter and 9.1 [5.7 to 20.6] % to 25.6 [14.2 to 50.0] % in summer, depending on the scenarios (Table E1). The changes simulated under the different scenarios show different amplitudes, but with similar patterns (Fig. 9).

### 5.2  Changes in the HMA region in the global context

HMA is projected to warm around 1.5 times faster than the atmosphere at the global scale, with a quasi-linear relationship between HMA and Global Surface Air Temperature (GSAT; Fig. 10a). This stronger signal over HMA is mainly explained



Earth System Dynamics Discussions — Open Access EGU

| | | Annual | | | | DJFMA | | | | JJAS | | | |
|---|---|---|---|---|---|---|---|---|---|---|---|---|---|
| | | HMA | HK | HM | TP | HMA | HK | HM | TP | HMA | HK | HM | TP |
| tas [°C] | ssp126 | 1.9 [1.2 2.7] | 1.9 [1.3 2.6] | 1.9 [1.0 2.5] | 1.9 [1.2 2.8] | 1.9 [1.0 2.7] | 1.8 [1.2 2.6] | 1.9 [0.9 2.6] | 1.9 [1.0 2.7] | 1.8 [1.1 3.0] | 1.8 [1.2 2.9] | 1.5 [1.0 2.5] | 1.8 [1.2 3.2] |
| | ssp245 | 3.4 [2.5 4.7] | 3.4 [2.6 4.7] | 3.4 [2.4 4.3] | 3.4 [2.5 4.8] | 3.4 [2.4 4.6] | 3.3 [2.4 4.6] | 3.5 [2.5 4.5] | 3.5 [2.5 4.5] | 3.0 [2.3 5.0] | 3.2 [2.4 5.0] | 2.7 [2.1 4.2] | 3.0 [2.3 5.3] |
| | ssp370 | 5.0 [3.8 7.3] | 5.1 [3.8 7.3] | 5.0 [3.7 6.8] | 5.0 [4.0 7.5] | 5.1 [3.9 7.1] | 4.9 [3.7 7.3] | 5.3 [4.1 7.1] | 5.1 [4.1 7.0] | 4.5 [3.3 7.5] | 4.8 [3.2 7.7] | 4.0 [3.0 6.6] | 4.5 [3.4 7.9] |
| | ssp585 | 6.5 [4.9 9.0] | 6.7 [5.1 9.0] | 6.5 [4.6 8.6] | 6.5 [5.0 9.3] | 6.8 [4.9 8.9] | 6.6 [4.8 8.9] | 7.0 [5.0 9.0] | 6.8 [5.0 8.8] | 5.8 [4.5 9.1] | 6.2 [4.8 9.4] | 5.1 [3.8 8.1] | 5.7 [4.4 9.5] |
| snc [%] | ssp126 | -4.4 [-10.0 -0.1] | -4.2 [-9.8 -1.5] | -4.4 [-10.2 0.3] | -4.5 [-10.2 0.3] | -3.2 [-9.7 2.0] | -1.9 [-7.0 0.3] | -3.3 [-11.4 2.0] | -2.7 [-8.8 3.2] | -2.8 [-14.6 -0.2] | -4.9 [-16.6 -0.3] | -2.3 [-14.6 0.1] | -3.0 [-16.2 -0.4] |
| | ssp245 | -7.8 [-15.4 -2.5] | -7.3 [-14.7 -3.5] | -8.7 [-16.8 -3.0] | -7.7 [-15.7 -1.9] | -6.8 [-15.3 0.0] | -4.3 [-12.0 -0.4] | -8.8 [-19.7 -1.5] | -5.6 [-12.5 1.5] | -4.7 [-21.2 -0.8] | -8.1 [-23.8 -0.6] | -4.0 [-21.6 -0.6] | -5.0 [-24.4 -1.1] |
| | ssp370 | -11.6 [-22.9 -4.8] | -11.1 [-21.8 -6.1] | -13.4 [-25.5 -6.4] | -11.4 [-23.5 -4.0] | -11.2 [-23.6 -2.8] | -7.5 [-20.0 -1.9] | -15.5 [-31.8 -4.3] | -9.0 [-19.3 -1.7] | -6.3 [-26.9 -1.1] | -11.3 [-31.1 -0.9] | -5.1 [-28.6 -0.9] | -6.7 [-30.5 -1.6] |
| | ssp585 | -14.5 [-27.4 -6.0] | -14.8 [-26.4 -8.6] | -16.4 [-30.6 -7.0] | -13.8 [-27.7 -5.3] | -14.6 [-30.4 -3.7] | -10.3 [-28.1 -2.7] | -19.4 [-38.8 -5.5] | -11.9 [-24.9 -2.6] | -7.2 [-29.1 -1.4] | -14.0 [-35.7 -1.5] | -5.4 [-31.9 -0.9] | -7.5 [-32.0 -2.0] |
| pr [mm.day-1] | ssp126 | 0.2 [0.0 0.5] | 0.1 [-0.1 0.5] | 0.3 [0.0 0.6] | 0.2 [0.0 0.4] | 0.1 [-0.1 0.3] | 0.2 [-0.2 0.5] | 0.1 [-0.2 0.3] | 0.1 [-0.0 0.2] | 0.3 [-0.1 0.9] | 0.1 [-0.1 0.5] | 0.6 [-0.1 1.6] | 0.3 [0.0 0.9] |
| | ssp245 | 0.3 [0.1 0.6] | 0.2 [-0.1 0.5] | 0.4 [0.0 0.8] | 0.3 [0.1 0.5] | 0.1 [-0.1 0.4] | 0.3 [-0.2 0.7] | 0.0 [-0.3 0.4] | 0.1 [0.0 0.2] | 0.5 [0.1 1.1] | 0.2 [-0.2 0.6] | 0.8 [0.2 2.0] | 0.5 [0.1 1.1] |
| | ssp370 | 0.4 [0.1 0.9] | 0.4 [-0.1 0.7] | 0.6 [0.1 1.5] | 0.4 [0.1 0.7] | 0.2 [-0.0 0.5] | 0.4 [-0.1 0.9] | 0.0 [-0.2 0.4] | 0.2 [0.1 0.3] | 0.6 [-0.0 1.8] | 0.3 [-0.4 1.0] | 1.3 [0.1 3.9] | 0.6 [0.1 1.4] |
| | ssp585 | 0.6 [0.2 1.2] | 0.4 [-0.0 0.8] | 1.0 [0.4 2.2] | 0.5 [0.2 0.9] | 0.3 [0.0 0.7] | 0.6 [-0.1 1.2] | 0.2 [-0.2 0.8] | 0.2 [0.1 0.5] | 0.8 [0.1 2.2] | 0.4 [-0.3 1.1] | 2.0 [0.6 4.9] | 0.8 [0.2 1.8] |

**Table 2.** Annual and seasonal (DJFMA and JJAS) multimodel median anomalies averaged over 2081-2100 (with respect to 1995-2014 average) and their 90 % confidence interval (0.05 and 0.95 quantiles) for temperature, snow cover and precipitation over the full HMA domain and the subregions HK, HM and TP, under the 4 SSPs scenarios: SSP1-2.6, SSP2-4.5, SSP3-7.0 and SSP5-8.5.





by the enhanced warming rates that occur over the continental areas with respect to oceanic regions, since the warming rates affecting the NH continental surfaces and the HMA are similar (Fig. 10b). Nevertheless, the warming rate is stronger over HMA than over the remaining continental areas located south of 60° N. This amplification with respect to other tropical to mid-latitude areas reaches 11 % and is significant at the 5 % level, and was already noticed in CMIP5 (Rangwala et al., 2013). This is potentially related to feedbacks involving snow cover changes, namely snow-albedo feedback, and other processes specific to mountainous areas. Mudryk et al. (2020) showed that the projected NH spring SCE decrease is proportional to the global temperature change, the slope of the relationship being independent of the scenario. Similarly, the HMA snow cover in DJFMA follows a linear decrease of about 4 % per degree of GSAT increase (Fig. 10d). This relationship does not show a linear behavior in summer, with a curve relationship highlighting small snow cover decrease rates at high levels of global warming (Fig. 10e). The summer curve relationship is explained by a fast retreat of snow cover at low elevation under moderate warming whereas some snow cover still persists at very high elevation even under strong warming rates. However, almost all the summer snow cover is vanishing in the highest $CO_2$ emissions scenario (SSP5-8.5). The annual precipitation increase also appears to behave linearly with the GSAT increase, with an increase of precipitation slightly higher than 6 % per degree of GSAT increase (Fig. 10f).

## 6 Discussion

The cold bias over the TP and mountainous areas has been a persistent issue in GCMs since the first AMIP experiments (Mao and Robock, 1998). The mean cold bias found in our study is coherent with previous CMIP5 studies as, for example, Su et al. (2013) that estimated a cold bias of 1.1 to 2.5 °C in winter and values not exceeding 1 °C in summer. Zhu and Yang (2020) showed an improvement from CMIP5 to CMIP6 with a mean bias reduction reaching 0.44 °C. However, the skill changes from CMIP5 to CMIP6 are contrasted among the models. MPI-ESM-LR was the model performing the best over HMA in terms of temperature RMSE in Su et al. (2013) Table 6 and it is still in the top 3 models in our study in his high-resolution version, while CanESM2 was in the top 5 models and dropped down to the 5 worst models in our study. Even if these results are not directly comparable because the studies do not focus on exactly the same periods, methods and area, this still gives an insight on the fact that not all models improved or maintained their ability to simulate temperature over HMA. Deteriorations might be due to some changes in the models. This is for example the case for IPSL-CM6A-LR which contains a new snow scheme (Wang et al., 2013a) that improved snow cover representation over most of NH but increased the biases over HMA (Cheruy et al., 2020).

Indeed, cold bias might be related to a misrepresentation of snow cover in some cases. As noted by Zhu and Yang (2020), the average model temperature bias is more pronounced in winter, suggesting a possible role of snow-albedo parameterizations. However, the bias correlation between temperature, snow cover and precipitation (Fig. 4) indicates that the temperature bias might not only be due to snow cover misrepresentation, in particular for some models that show low correlations between temperature and snow cover (e.g. CAS-ESM-2-0, CESM2) whereas some of them have a high correlation with precipitation. However, all these variables are connected, as an excess of precipitation can induce an overestimation of snow cover and gener-





ate a cold bias, but an initial cold bias in the atmosphere could also generate an overestimation of solid precipitation leading to an excess of snow cover. Nevertheless, the correlation between snow cover bias and precipitation bias is less obvious for most of the models. Exceptions are BCC-ESM1 and CAS-ESM2-0, where these variable biases are highly correlated (0.48 and 0.41), indicating in that case that precipitation might be partly the cause of the temperature biases. However, models that display the

strongest cold biases, such as CNRM-CM6-1-HR, IPSL-CM6A-LR and CNRM-CM6-1, also show high anticorrelation with snow cover bias (-0.39, -0.62 and -0.5 respectively). This supports the recommendation by Zhu and Yang (2020) to improve snow cover parameterizations in models. In addition, as mentioned by Gu et al. (2012), the lack of high-elevation observation stations in the CRU data may also be partly responsible for the apparent cold bias of the models. Direct comparison from CRU and models can also amplify this bias, knowing that differences of elevation can be noted between GCMs. Therefore, certain

studies (e.g., Sheffield et al., 2006; Chen et al., 2017) correct temperatures with a common lapse rate (e.g. 6.5 °C km$^{-1}$) to bring them to the same elevation. However, in our case where we wanted to correlate the model biases of different variables and see the impact of original model resolution, this method would have introduced additional uncertainties due to spatially heterogeneous lapse rates over this region and partially corrected the biases due to model resolution, making it difficult to compare the variables with each other.

Other variables as cloud cover and aerosols might also be important factors involved in model biases. Indeed, the aerosol effect on snow cover is often investigated. It induces a reduction of the seasonal snow cover duration of a few days, in particular over HMA (Ménégoz et al., 2014), but regarding the large snow cover biases in models, it might be a second order issue for highest biased models. However, the spatial imprint of aerosol forcing seems highly correlated with snow cover biases (see Usha et al. (2020) Figure 7d and Fig. B1 of this paper). Usha et al. (2020) conclude that snow darkening due to aerosols

increases the surface temperature by 1.33 ± 1.2 K, which results in the reduction of snow cover fraction by 7 ± 11 %. Therefore a misrepresentation in aerosols deposition on snow might amplify snow cover biases and/or be the main cause for some models, while the multimodel mean absolute snow cover bias is estimated to 12 [−13 to 43] % in this study. Zhou and Li (2002) and Yu et al. (2004) conjectured that a poor representation of cloud properties might lead to insufficient plateau heating, resulting in a cold bias in the TP. The problem could also arise from a misrepresentation of processes in the boundary

layer. Indeed, De Wekker and Kossmann (2015) and Serafin et al. (2020) expose the lack of constraints for processes in the planetary boundary layer over complex terrain, in addition to the limited applicability of existing turbulence theory with frequent violation of its basic assumptions (e.g. stationarity and isotropy of small-scale turbulence) over mountainous areas. Further theoretical and observation work is thus needed to improve model parameterizations over these regions. Chen et al. (2017) focused on the surface energy budget from 28 CMIP5 models, and suggested that improvements in the parameterization

of the snow cover area and the boundary layer processes should allow reducing the cold bias over the TP. Further research is required to advance scientific understanding about the origins of systematic model biases in the HMA region.

    Some models also show temperature biases in the troposphere at a global scale (not shown) that might amplify and/or even trigger surface biases in HMA. Salunke et al. (2019) also highlighted that many models still struggle to capture the large-scale atmospheric circulation, such as the location and intensity of upper-level Asian anticyclone and middle troposphere

temperature maximum over the TP, which have large implications on the TP as well as on the Indian summer monsoon. Wrong



atmospheric circulation, as the position of jets, could also feed the observed biases in models over HMA. Further analyses in higher atmospheres and circulation must be done to quantify this impact on present biases.

The obvious link between topography, snow cover bias and to a lesser extent the temperature bias (Fig. 4), indicates also the inability of GCMs to simulate key variables over complex topography. Furthermore, increasing the model resolution does not
result in a systematic reduction of model biases (Fig. 5b, d, f) because resolutions of 50 km or higher are still too coarse to represent adequately all the physical processes peculiar to the complex topography of the HMA region. Hence, the development of subgrid-scale parameterizations appears essential to better simulate the climate of these regions. As an example, the snow cover extent parameterization is often too simple in actual GCMs. Many GCMs either use a simple linear relationship with snow water equivalent (SWE), as in the ERA-Interim reanalysis, or take also into account a dependency to the snow density as
implemented in ERA5 and IPSL-CM6A-LR for example. However, many models do not include any representation of subgrid-scale processes driven by the local topography, an essential feature for complex topography areas. This idea has been explored a few decades ago (e.g., Walland and Simmonds, 1996; Roesch et al., 2001) and more recently in Swenson and Lawrence (2012) who considered satellite observations to develop a subgrid-scale representation of snow cover with a dependency on the local topography. Such approaches are however challenged by the lack of spatialized observation of SWE over mountainous
areas at a global scale.

Regarding the disparity in observed past snow cover trends, it has to be noted that the NOAA CDR product is a binary product at coarse resolution (∼200 km) based on satellite observations and new satellite has allowed more accurate observation of HMA over the past decades (e.g. Meteosat-5; Helfrich et al. (2007)), in addition to a change from manual to automatic charts and increasing observational resolution from IMS (see Sect. 2.2.2). Trends apparent in the NOAA CDR, therefore, need to be
taken with caution, above all for mountain regions (Bormann et al., 2018). Figure D2 compares the trends from NOAA CDR, ERA-Interim and ERA5, and shows spatial discrepancies. However, ERA-Interim as ERA5 trends can also be affected by changes in observational assimilation as, for example, the inclusion of IMS snow assimilation in 2004 (Drusch et al., 2004; Hersbach et al., 2020). It is therefore not easy to conclude whether the observed trends are due to a part of internal variability, which would not be found in the multimodel mean, or to artifacts of the observation products themselves. Past studies on snow
cover trends show contradictory results depending on the study zone, methods and period (e.g., Dahe et al., 2006; Pu et al., 2007; Immerzeel et al., 2009; Shen et al., 2015; Notarnicola, 2020). More recently Li et al. (2018) showed a slight snow cover decrease of about 1.1 % during 2001–2014 over TP with high-resolution MODIS product, which seems in closer agreement with the large scale multimodel mean snow cover trends displayed on Fig. 6f, h.

The low resolution of the NOAA CDR snow cover product and the fact that it only provides binary values might also impact
the spatial distribution of biases by overestimating snow cover where large snow amounts can be found and underestimate it where low values of snow cover are found. However, this effect should mostly be reduced by spatial averages. The recently published ESA CCI snow cover product at a better temporal and spatial resolution, even if not exactly on the same period, suggests that snow cover biases in models can even be higher over HMA. Generally, the complex and highly variable snow cover behaviour calls for better spatio-temporal resolution than currently available from present-day snow cover products based
on both observations, reanalyses or models and thus hampers reliable trend detection.



Precipitation is particularly uncertain over HMA. It is well known that most of the observational datasets underestimate solid precipitation (Palazzi et al., 2013; Sanjay et al., 2017; Sun et al., 2018) due to gauge undercatch. An additional issue is the scarce observational network in this region. Using glacier mass balances to infer the high-altitude precipitation in the upper Indus basin, Immerzeel et al. (2015) suggests an underestimation of precipitation reaching a factor of two to ten in observational

datasets. Most studies comparing models and reanalyses with observational datasets do also account for large differences from about 100 % to 200 % (e.g., Palazzi et al., 2013; Su et al., 2013; Salunke et al., 2019). GPCP is in closer agreement with CMIP6 models, which could be explained by a better representation of solid precipitation. Overall, it is challenging to estimate model biases, even if the coarse GCMs resolution might underestimate the orographic barrier effect of high mountains leading to a potential excess of water vapor transport toward the TP (Lin et al., 2018). Hence, all models and analyses regarding

precipitation need to be considered with caution.

On one hand, a general increase of precipitation and a general decrease of snow cover is expected over most of HMA as a response to anthropogenic forcings in future projections (Sabin et al., 2020). On the other hand, past precipitation trends are heterogeneous depending on the location, the season and the period (Yoon et al., 2019), a result also found for snow cover (e.g., Li et al., 2018). A snow cover decrease located mainly in the west border of HK and in the southeast of TP including HM in

winter is expected in future projections (Fig. 9e, f), whereas small changes are expected for this variable in the center of the HK region, in particular over the Karakoram, even under high $CO_2$ scenarios emissions. The glacier growth occurring in this area over the last decades, defined as the Karakoram anomaly (e.g., Brun et al., 2017), might be associated with an intensification of WDs (Krishnan et al., 2019; Sabin et al., 2020), a finding confirmed in our study with the increase in winter precipitation found in the projections (Fig. 9i, j) from 5.6 [−1.9 to 18.1] % to 19.9 [5.3 to 54.1] % depending on the SSPs (Table E1).

Indeed the increase in temperature might be offset by the increase of winter snowfall over the Karakoram slowing down the decrease of snow cover in this area. Concerning the summer precipitation, a declining trend of the Indian summer monsoon precipitation during the post-1950 has been reported (Krishnan et al., 2013, 2016; Sabin et al., 2020), while CMIP6 analyses are showing the opposite under future warming pathways (Katzenberger et al., 2021) with an intensification of precipitations by most of CMIP6 models, that our results also confirm with a precipitations increase over most of TP and HM (Fig. 9i, j)

with a relative increase over HM of 7.9 [5.3 19.9] % to 28.5 [13.9 to 78.7] % (Table E1). The apparent contradiction between past and future trends is likely related to aerosol forcing that shows a strong variability both spatially and temporally. Das et al. (2020) suggest an ongoing decrease of the black and organic carbon atmospheric loads associated with an increase in sulfate concentration over the Indian subcontinent. This leads to a general increase in the aerosol depth that counteracts the warming related to greenhouse gases (GHGs) through a scattering of the solar radiation, but with a more heterogeneous spatial pattern

that modulates precipitation changes at the local scale. If the aerosol forcing plays a major role at present and over the near future, the GHG forcing is expected to dominate at the end of the XXIth century, leading to a general intensification of the hydrological cycle and a strengthening of the Indian summer monsoon.

Recent studies suggest an overestimation of the warming rates in CMIP6 models (Forster et al., 2020). This could be partly related to an overestimation of the climate sensitivity in this new model generation, probably related to cloud and cloud-aerosol

schemes (Meehl et al., 2020). This question is challenged by the cooling effect of the aerosols that is expected to decrease in



relation to the ongoing improvements of air quality, potentially increasing the warming rates at the regional scale (Turnock et al., 2020). An open question is the possibility to exclude models that simulate past trends incompatible with the observed ones, a way to reduce the uncertainties in future projections by excluding, in particular, the models that go beyond or below realistic warming rates (e.g., Ribes et al., 2021). This approach is relevant at the global scale, but more challenging at the
regional scale, where multi-decadal trends can originate from internal variability (Hawkins et al., 2016), a spatial scale for which aerosol signals might also play a major signal, in particular over polluted areas. In any case, the models considered in the present study show ensemble trends that encompass the observed past trends of temperature, precipitation and snow cover over 1979-2014, except for one model for the temperature (Fig. 7a, d). It is important to remember that even some of the strongly biased models are able to reproduce the past trends, suggesting that a simple bias analysis would not be sufficient to
select a subset of CMIP models for climate applications. This result might justify keeping all models for projections supposing that their biases are stationary, a hypothesis that has already been shown in a modeling study (Krinner et al., 2020).

## 7   Conclusions

In this study, we assessed the performances of 26 CMIP6 GCMs over HMA for the historical period 1979-2014 and the future projections from 10 of them under the four shared socioeconomic pathways SSP1-2.6, SSP2-4.5, SSP3-7.0 and SSP5-
8.5, through 3 variables: near-surface air temperature, snow cover extent and total precipitation. Cold bias over HMA is still present in this latest generation of GCMs with an average annual underestimation of $-1.9$ [$-8.2$ to 2.9] °C compared to the CRU dataset, associated with an average snow cover overestimation of 12 [$-13$ to 43] % (or 52 [$-53$ to 183] % relatively) compared to the NOAA CDR satellite observation. The recently published ESA CCI snow cover product at a better temporal and spatial resolution shows lower snow cover values compared to NOAA CDR, suggesting that model biases may be even
larger. The temperature and snow cover model biases are more pronounced in winter. Precipitation is also overestimated by 1.5 [0.3 to 2.9] mm d$^{-1}$ (or 143 [31 to 281] % relative) but this later difference might mostly reflect the undercatch of solid precipitation in APHRODITE.

   For most models, the cold surface bias is associated with an overestimation of snow cover, but this is not the case for all models. The snow cover bias originates from precipitation biases in some models, with large amounts of snowfall significantly
affecting the snow cover. Therefore, the source of biases might strongly differ from one model to another. The study of temperature, snow cover and precipitation can only partly explain these biases which might be affected by other factors including cloud cover, aerosols and atmospheric circulation. Some models also show temperature biases in the troposphere that might trigger or amplify surface biases. Hence, further analyses focusing on the higher atmosphere are required. All models also show a significant correlation between snow cover bias and surface elevation (and to a lesser extent between temperature bias and elevation), underlining the challenge to correctly simulate snow cover at high elevation. Besides, best performing models
for temperature are not necessarily the same for the other variables and increasing the model resolution does not show any clear improvement over this region, suggesting that further work on theory and model parameterization is essential. In addition, a dependency of the simulated past trends to the model biases is found for some variables and seasons, however, some highly



biased models fall within the range of observed trends, suggesting that model bias is not a robust criterion to discard models in
trend analysis.

The 10 models, used with future scenarios, project a HMA median warming at the end of this century (2081-2100 with
respect to 1995-2014 average) that ranges from 1.9 [1.2 to 2.7] °C for SSP1-2.6 to 6.5 [4.9 to 9.0] °C for SSP5-8.5. The overall
warming is associated with a relative median snow cover decrease from $-9.4$ [$-16.4$ to $-5.0$] % to $-32.2$ [$-49.1$ to $-25.0$] %
and a relative median precipitation increase from 8.5 [4.8 to 18.2] % to 24.9 [14.4 to 48.1] % by the end of the century. The
warming over HMA is projected to be 11 % stronger than over the mean of NH continental surfaces, excluding the Arctic
domain. The HMA projected temperature, snow cover and precipitation show a linear relationship with the GSAT increase,
except for summer snow cover that shows slower melt rates for high temperature levels.

Further work is required to build realistic climate models over HMA domain. This task is challenging because it requires
model parameterizations adapted to high elevation areas that might require high resolution. These models should be able also
to simulate the response of the climate system to GHGs and to the regional imprint of the aerosol forcing, an essential feature
in this strongly polluted area. Finally, such models should be also global, since the HMA area is a major piece of the climate
system with climate teleconnections found all over the world.

*Code availability.* All scripts to produce the figures and results are available at: https://github.com/mickaellalande/PhD/tree/master/CICLAD/
Himalaya/CMIP6_HMA_paper. We use Python (Oliphant, 2007; Millman and Aivazis, 2011) version 3.8.5 and xarray (Hoyer and Hamman,
2017) version 0.16.0 to manipulate netCDF files. Interpolations are performed using xESMF version 0.3.0 (https://doi.org/10.5281/zenodo.
1134365). For statistical purposes, Scipy (Virtanen et al., 2020) version 1.5.2 is used. All graphics are made using Proplot version 0.6.4 based
on Matplotlib (Hunter, 2007) version 3.2.2 and Cartopy version 0.18.0.

*Data availability.* The datasets from CMIP6 simulations are available via the CMIP6 Search Interface: https://esgf-node.llnl.gov/search/
cmip6/. CRU TS (Climatic Research Unit gridded Time Series) version 4.00 is available at http://doi.org/10/gbr3nj. NOAA Climate Data
Record (CDR) of Northern Hemisphere (NH) Snow Cover Extent (SCE), Version 1: https://doi.org/10.7289/V5N014G9. The ESA CCI
snow product is available here: https://catalogue.ceda.ac.uk/uuid/5484dc1392bc43c1ace73ba38a22ac56 and the cloud gap filter used in this
study can be provided upon request. The APHRODITE products are available here: http://aphrodite.st.hirosaki-u.ac.jp/download/. Global
Precipitation Climatology Project (GPCP) Climate Data Record (CDR), Version 2.3 (Monthly): https://doi.org/10.7289/V56971M6. Global
Multi-resolution Terrain Elevation Data 2010 (GMTED2010): https://www.temis.nl/data/gmted2010/index.php. The GLDAS Land/Sea Mask
Dataset at 1° can be found here: https://ldas.gsfc.nasa.gov/gldas/vegetation-class-mask. We also used the ERA-Interim and ERA5 reanalyses
data that are publicly available at: https://www.ecmwf.int/en/forecasts/datasets/reanalysis-datasets/era-interim and https://www.ecmwf.int/
en/forecasts/datasets/reanalysis-datasets/era5.



**Figure 3.** Annual bias (model minus observation) computed over 1979-2014 for temperature, except the top left panel that shows the climatology estimated from the CRU observation, used as the reference for the bias computation. The panel located at the right side of the CRU observation shows the bias of the multimodel mean based on the 26 models shown in the figure. The black contour shows the political frontiers and the bold black line the HMA domain located above 2500 m asl for which is given the spatial average of the bias at the bottom left of each panel.



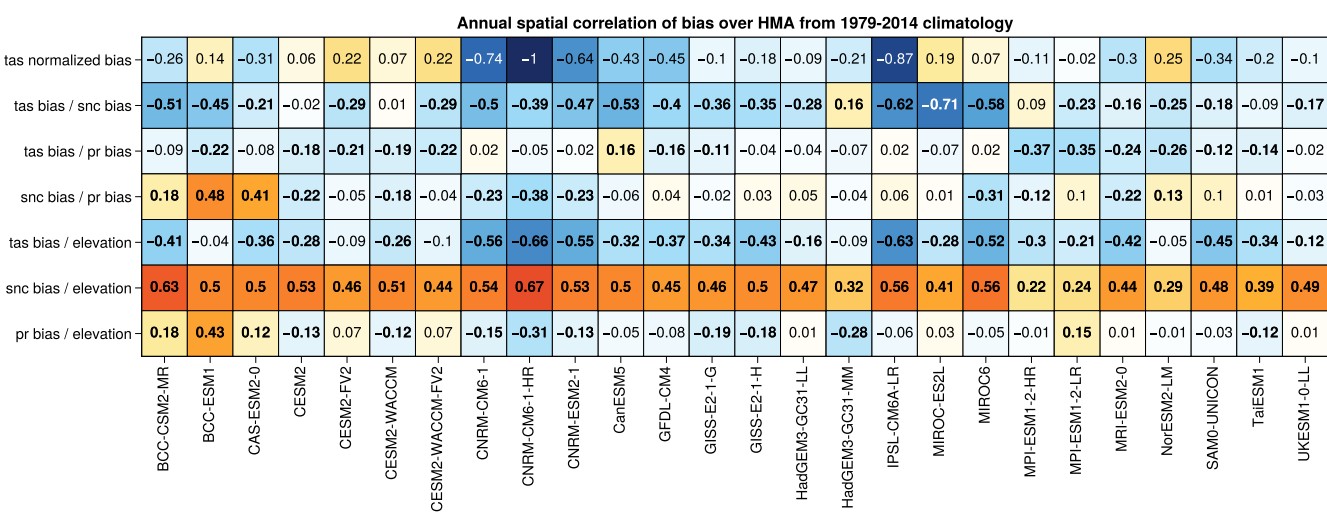

**Figure 4.** Pattern correlations of the annual model biases. The first row shows the temperature bias normalized with the strongest temperature bias found among the 26 models (CNRM-CM6-1-HR). The following rows show the pattern correlations computed between temperature and snow cover biases (second row), temperature and precipitation biases (third row), snow cover and precipitation biases (fourth row). The fifth to the seventh rows show the correlation between biases and surface elevation estimated from GMTED2010 for temperature, snow cover and precipitation. All biases are annual and computed over 1979-2014. Bold characters highlight significant correlation (p-value < 0.05).



**Figure 5.** Annual spatial RMSE and bias computed over HMA with respect to CRU, NOAA CDR and APHRODITE, respectively for temperature, snow cover and precipitation (a, c, e). Models are ranked by increasing RMSE for temperature and the multimodel mean appears in the first histogram. The approximate original model's resolution is given, but all metrics are computed on a common 1° x 1°grid after interpolation. (b, d ,f) are similar to (a, c, e) with models ranked as a function of their original mean lat/lon resolution. Blue and red crosses correspond respectively to RMSE and mean bias.



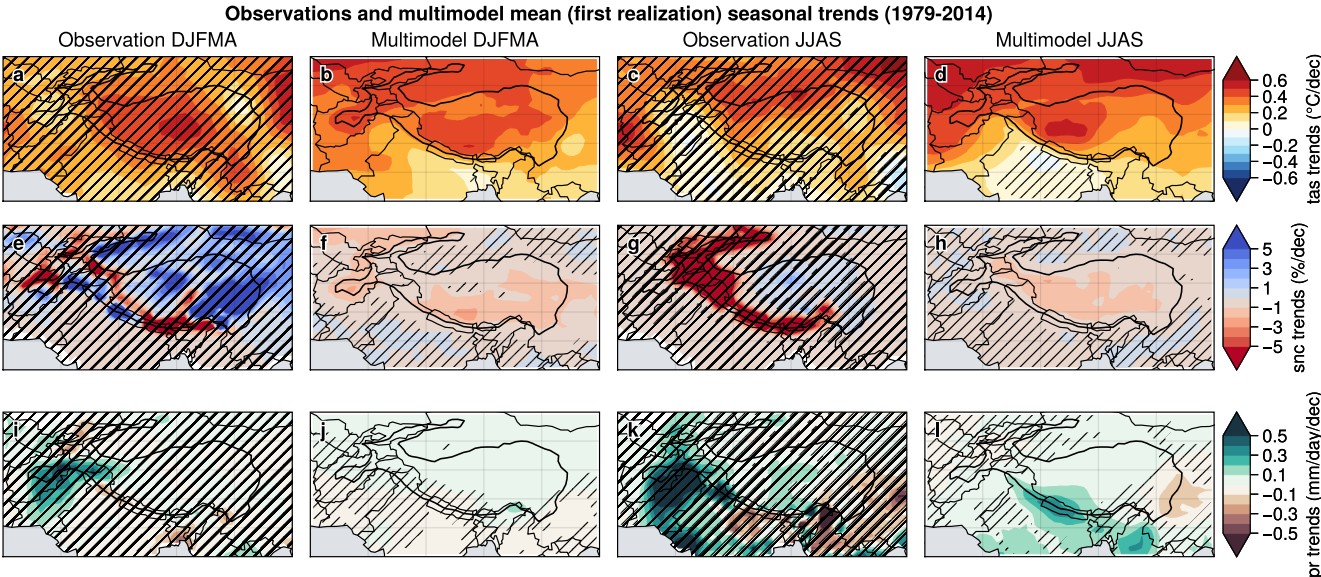

**Figure 6.** DJFMA (left) and JJAS (right) trends computed over 1979-2014 for temperature (a-d), snow cover (e-h) and precipitation (i-l). CRU temperature, NOAA CDR snow cover and APHRODITE precipitation observations trends (DJFMA: a, e, i and JJAS: c, g, k) are compared to the multimodel mean computed with the first realization for each model (DJFMA: b, f, j and JJAS: d, h, l). Hatched areas highlight the regions where the trend is not significant (p-value > 0.05).

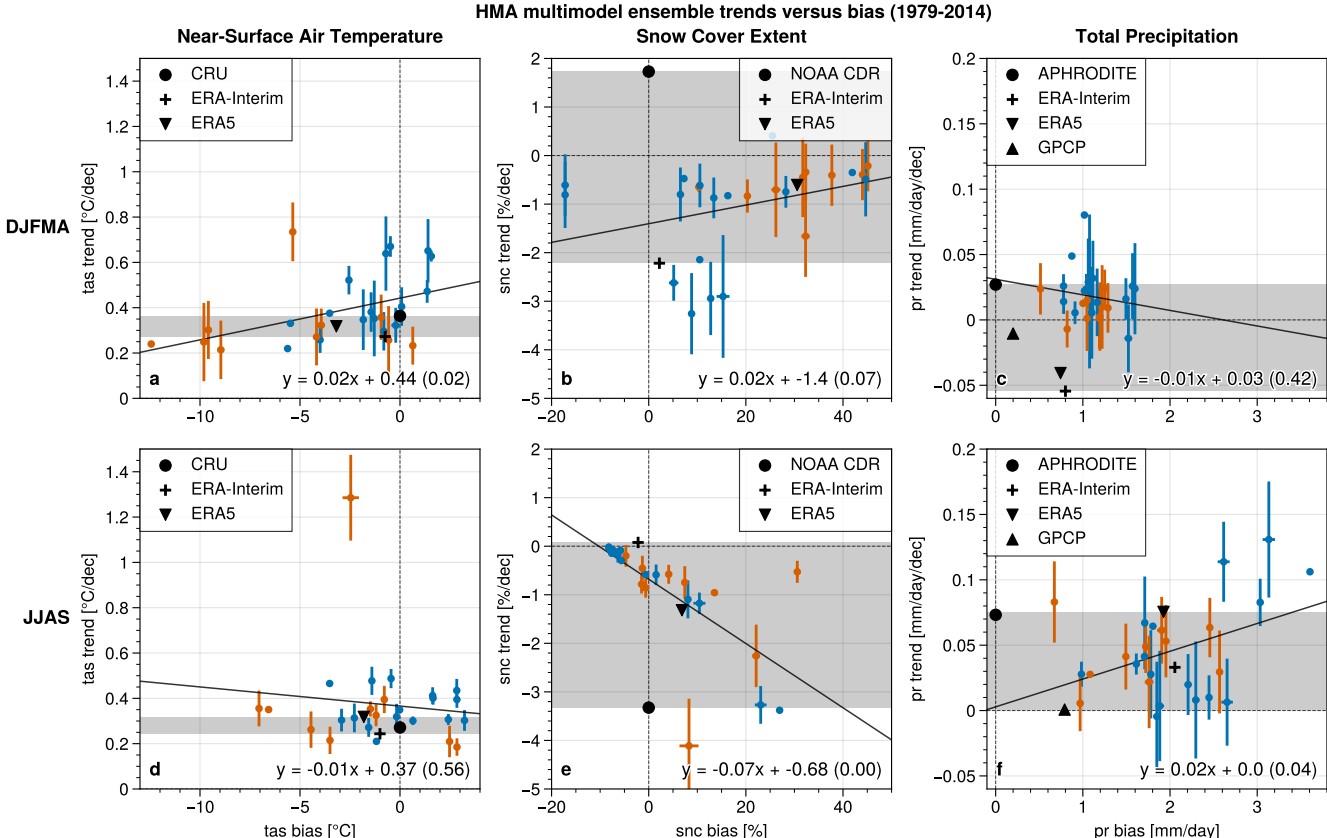

**Figure 7.** Seasonal (DJFMA and JJAS) HMA multimodel ensemble trends versus model biases over 1979-2014 for temperature (a, d), snow cover (b, e) and precipitation (c, f). Vertical (horizontal) bars correspond to the standard deviation of the trends (biases) of the ensemble members for each model (the number of members differs among the models Table A1). Observation and reanalysis datasets are shown with black symbols. The 10 models used for projections are shown in orange while other models are shown in blue. The grey shading represents the range of observation trends. The black solid line corresponds to a linear regression from the multimodel mean values (including all models) with the p-value shown in brackets following the equation at the bottom right of each panel. The light horizontal (vertical) dashed lines correspond to a null trend (null bias with respect to the reference observation dataset).

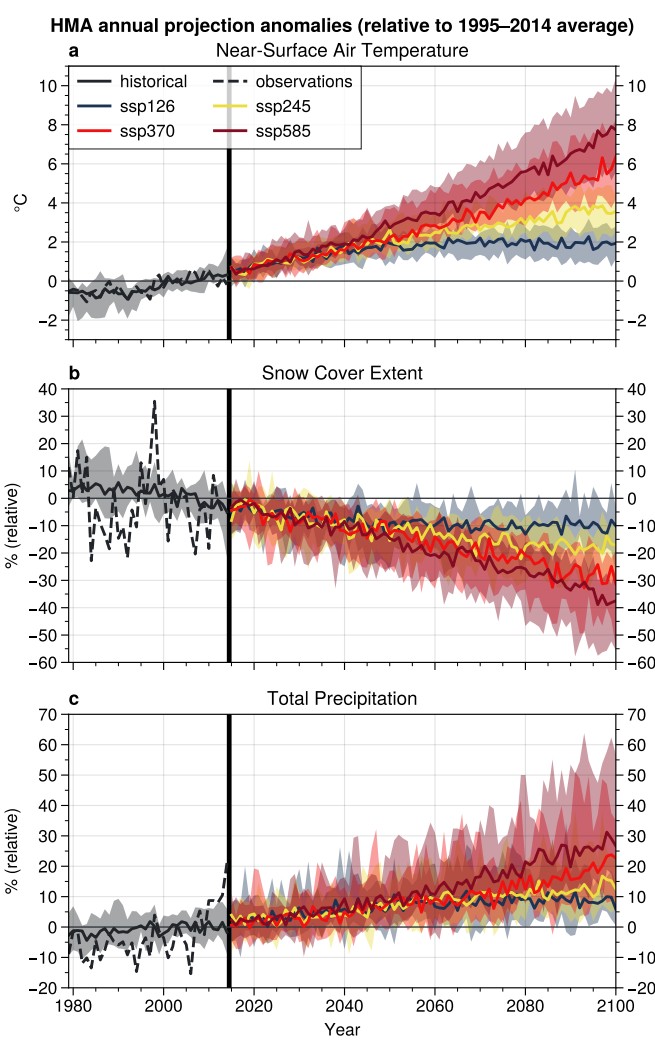

**Figure 8.** Annual yearly anomalies (with respect to the mean computed over 1995-2014) simulated in HMA over 1979-2014 (black curves) and over 2015-2100 under 4 scenarios: SSP1-2.6 (dark blue), SSP2-4.5 (yellow), SSP3-7.0 (red) and SSP5-8.5 (dark red) for temperature (a), snow cover (b) and precipitation (c). Medians are computed with the first member for the 10 models for which the future projections are available (Table 1). The thick vertical black line delimits the historical and future periods. CRU, NOAA CDR and APHRODITE observations dataset are shown for the historical period in dashed lines, respectively for temperature, snow cover and precipitation. Shadings highlight the 90 % confidence interval, corresponding to the 0.05 and 0.95 quantiles.

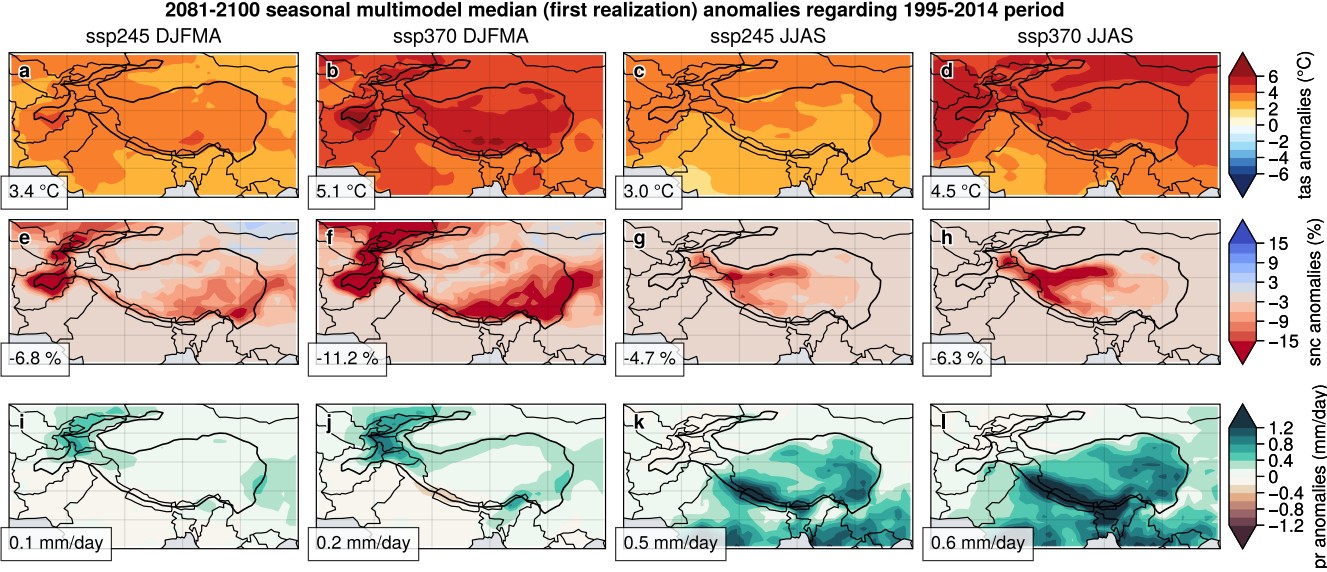

**Figure 9.** Median of the model differences between the 2081-2010 and the 1995-2014 averages for temperature (a-d), snow cover (e-h) and precipitation (i-l) under the scenarios SSP2-4.5 and SSP3-7.0. The first realization is used for each model. The black contour corresponds to the HMA domain (> 2500m), for which the spatial average is shown in the lower-left box.

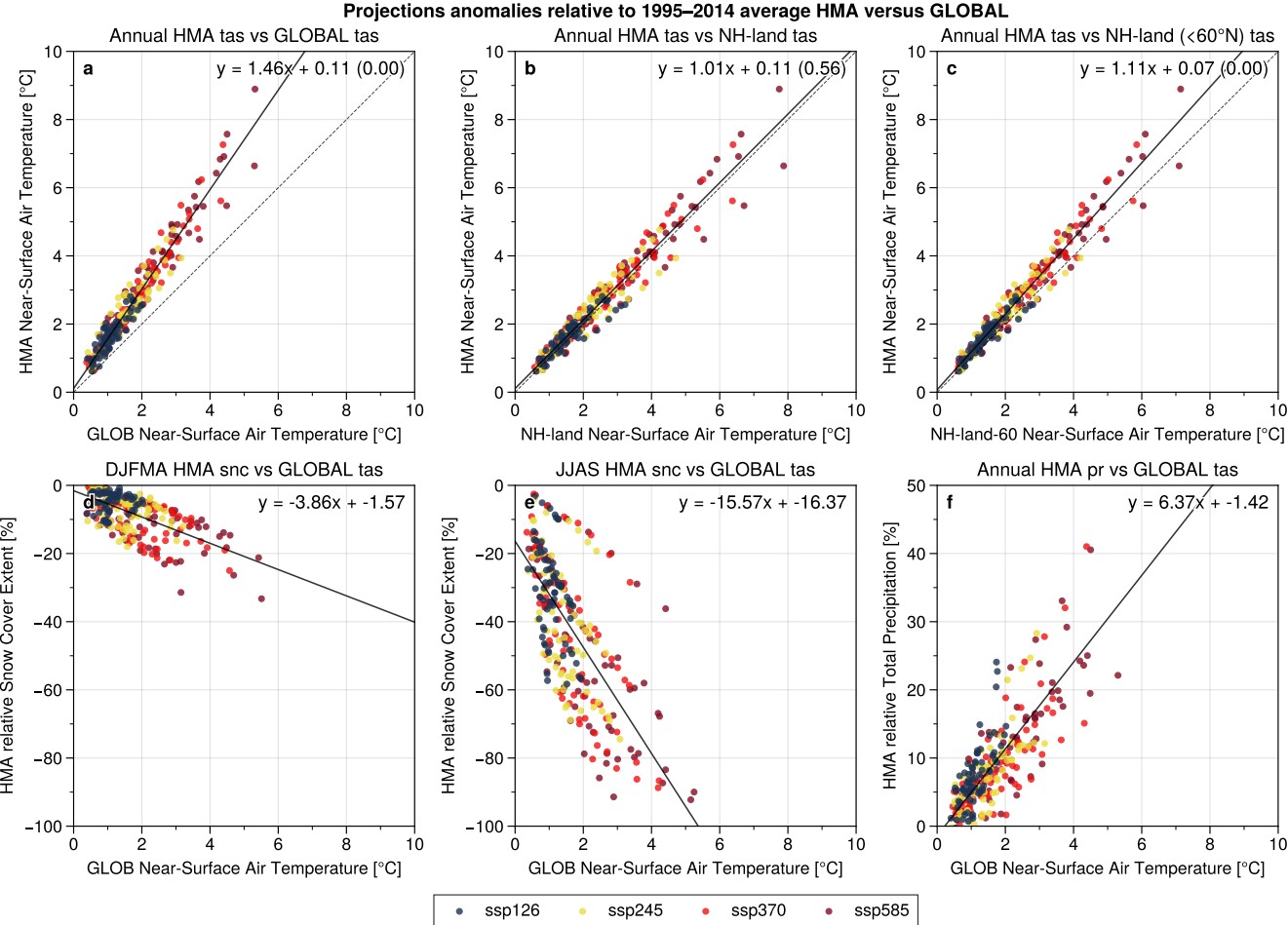

**Figure 10.** Projected anomalies over HMA computed with respect to 1995-2014 averages for the 10 models under the scenarios SSP1-2.6 (dark blue), SSP2-4.5 (yellow), SSP3-7.0 (red) and SSP5-8.5 (dark red) for annual temperature (a, b, c), seasonal snow cover (d, e) and annual precipitation (f) as a function of temperature integrated globally (a, d, e, f) and over NH continental surfaces including (b) and excluding (c) boreal areas (latitude > 60° N). Each point on the scatter plots corresponds to a 20-year average computed with 10-year steps from 2015 to 2095. The GLDAS Land/Sea Mask Dataset at 1° resolution (https://ldas.gsfc.nasa.gov/gldas/vegetation-class-mask) is used to mask the continental data in the different models. The dashed line corresponds to the line of equation $y = x$ and the solid black line to a linear regression detailed at the top right of each panel. In the first row (a-c), a significance Wald Test is done to test if the observed trend is significantly different from the equation $y = x$ (p-value is shown in brackets).





## Appendix A:  Members for trend analysis

| CMIP6 institute | CMIP6 model | Members for trend analysis | | |
| --- | --- | --- | --- | --- |
| | | tas | snc | pr |
| BCC | BCC-CSM2-MR | r1-3i1p1f1 (3) | | |
| | BCC-ESM1 | | | |
| CAS | CAS-ESM2-0 | r1-4i1p1f1 (4) | | |
| NCAR | CESM2 | r1-11i1p1f1 (11) | | |
| | CESM2-FV2 | r1-3i1p1f1 (3) | | |
| | CESM2-WACCM | | | |
| | CESM2-WACCM-FV2 | | | |
| CNRM-CERFACS | CNRM-CM6-1 | r1-30i1p1f2 (30) | | r1-29i1p1f2 (29) |
| | CNRM-CM6-1-HR | r1i1p1f2 (1) | | |
| | CNRM-ESM2-1 | r1-5,7-11i1p1f2 (10) | r1-6,8-11i1p1f2 (10) | r1-5,7-11i1p1f2 (10) |
| CCCma | CanESM5 | r1-40i1p2f1 (40) | | |
| NOAA-GFDL | GFDL-CM4 | r1i1p1f1 (1) | | |
| NASA-GISS[2] | GISS-E2-1-G | r1-10i1p1f1-2 (20) | | |
| | GISS-E2-1-H | r1-5i1p1f1, r1-10i1p1f2 (15) | | |
| MOHC | HadGEM3-GC31-LL | r1-4i1p1f3 (4) | | |
| | HadGEM3-GC31-MM | | | |
| IPSL | IPSL-CM6A-LR | r1-32i1p1f1 (32) | | |
| MIROC | MIROC-ES2L | r1-10i1p1f2 (10) | | |
| | MIROC6 | r1-50i1p1f1 (50) | | |
| MPI-M | MPI-ESM1-2-HR | r1-10i1p1f1 (10) | | |
| | MPI-ESM1-2-LR | | | |
| MRI | MRI-ESM2-0 | r1-5i1p1f1 (5) | | |
| NCC | NorESM2-LM | r1-3i1p1f1 (3) | | |
| SNU | SAM0-UNICON | r1i1p1f1 (1) | | |
| AS-RCEC | TaiESM1 | r1i1p1f1 (1) | | |
| MOHC, NIMS-KMA | UKESM1-0-LL | r1-4,8-12,16-19i1p1f2 (13) | r1-4,8-10,16-19i1p1f2 (11) | r1-4,8-12,16-19i1p1f2 (13) |

**Table A1.** Details of the members used from CMIP6 models for trends analyses for each variable.

**Figure B1.** Same as Fig. 3 for snow cover extent.

## Appendix B: Snow cover and precipitation biases





**Figure B2.** Same as Fig. 3 for total precipitation.





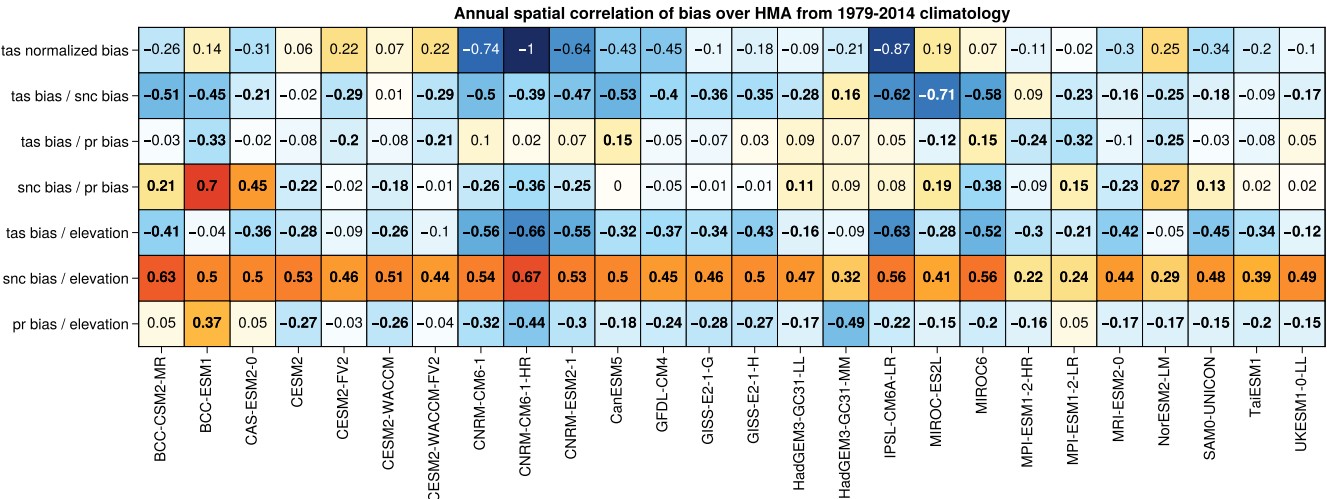

**Figure C1.** Same as Fig. 4, but with GPCP total precipitation observations instead of APHRODITE.

**Appendix C: Spatial bias correlation**





**Figure C2.** Same as Fig. 4, but for HK region and seasonal analysis.



**Spatial correlation of bias over TP from 1979-2014 climatology**

**Figure C3.** Same as Fig. 4, but for TP region and seasonal analysis.



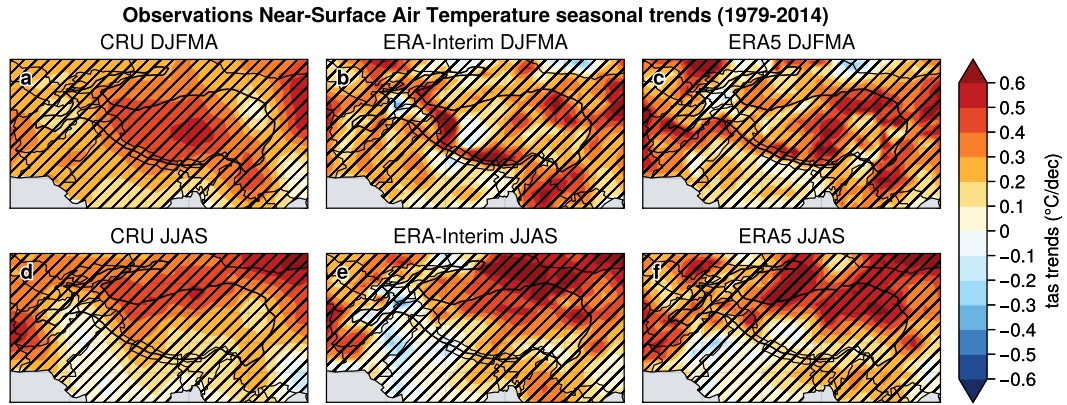

**Figure D1.** Same as Fig. 6 but for observations and reanalyses comparison with only near-surface air temperature.

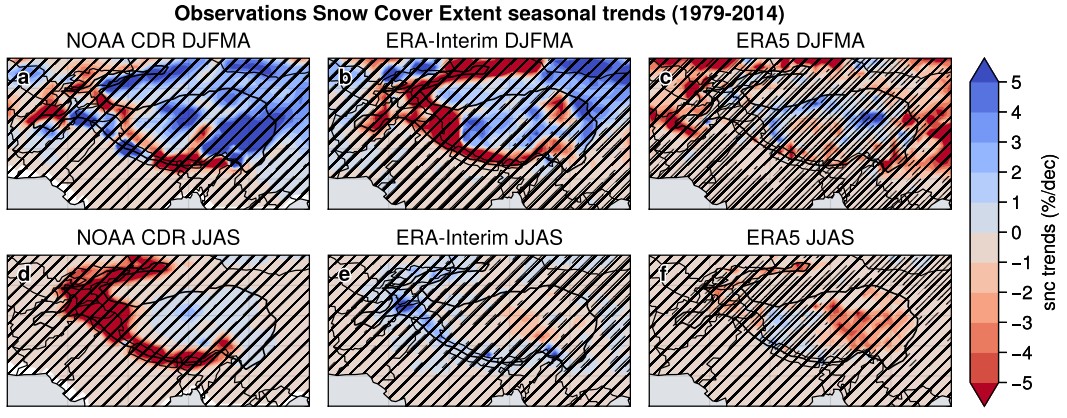

**Figure D2.** Same as Fig. 6 but for observations and reanalyses comparison with only snow cover extent.

**Appendix D:  Comparison of observations and reanalyses trends**





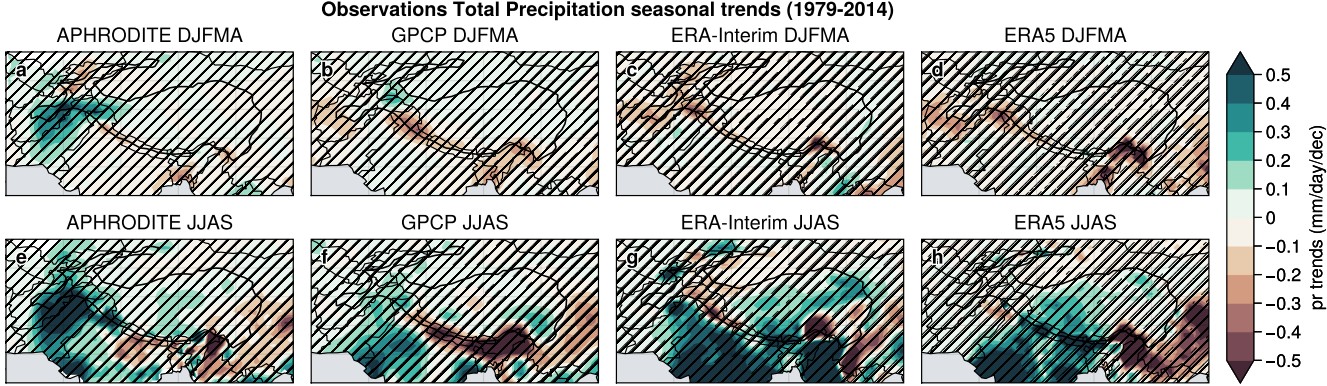

**Figure D3.** Same as Fig. 6 but for observations and reanalyses comparison with only total precipitation.



## Appendix E: Projected relative changes over HMA




|  |  | Annual | | | | DJFMA | | | | JJAS | | | |
|---|---|---|---|---|---|---|---|---|---|---|---|---|---|
|  |  | HMA | HK | HM | TP | HMA | HK | HM | TP | HMA | HK | HM | TP |
| snc [%] | ssp126 | -9.4 [-16.4 -5.0] | -8.0 [-13.0 -4.5] | -13.4 [-20.6 -3.4] | -8.4 [-16.0 -4.2] | -4.6 [-8.3 -2.3] | -2.9 [-6.7 -1.1] | -6.6 [-13.4 0.0] | -3.4 [-8.2 -1.5] | -32.2 [-54.9 -13.2] | -24.3 [-64.2 -10.0] | -42.2 [-61.4 -6.0] | -35.9 [-54.8 -14.0] |
|  | ssp245 | -17.8 [-26.2 -12.9] | -12.1 [-21.9 -7.8] | -24.9 [-34.9 -15.4] | -17.2 [-25.5 -12.0] | -9.5 [-15.6 -6.9] | -5.3 [-14.0 -2.2] | -15.3 [-29.1 -8.0] | -7.2 [-14.2 -4.7] | -53.0 [-72.3 -31.5] | -39.2 [-77.7 -19.0] | -67.1 [-87.0 -40.3] | -58.3 [-71.7 -36.1] |
|  | ssp370 | -25.7 [-41.9 -20.5] | -18.4 [-32.9 -12.3] | -36.7 [-54.3 -27.0] | -24.6 [-40.5 -18.5] | -17.1 [-27.8 -10.7] | -9.2 [-22.6 -4.3] | -27.8 [-48.9 -12.5] | -13.8 [-23.1 -7.4] | -69.8 [-90.7 -42.7] | -58.9 [-94.8 -25.6] | -84.2 [-98.0 -60.5] | -76.1 [-90.3 -49.1] |
|  | ssp585 | -32.2 [-49.1 -25.0] | -25.1 [-43.7 -16.6] | -40.7 [-65.2 -32.3] | -30.4 [-46.8 -22.1] | -21.8 [-36.8 -14.8] | -12.4 [-33.3 -6.6] | -28.7 [-61.3 -19.1] | -20.1 [-28.8 -9.4] | -80.7 [-94.2 -51.5] | -75.0 [-97.9 -34.0] | -90.4 [-99.0 -70.8] | -85.5 [-93.7 -58.1] |
| pr [%] | ssp126 | 8.5 [4.8 18.2] | 4.8 [1.8 15.9] | 6.6 [4.4 15.7] | 11.9 [5.4 23.2] | 6.4 [0.7 13.5] | 5.6 [-1.9 18.1] | 2.9 [-3.6 8.6] | 8.4 [5.8 22.4] | 9.1 [5.7 20.6] | 8.8 [1.6 16.1] | 7.9 [5.3 19.9] | 11.0 [4.8 25.8] |
|  | ssp245 | 12.9 [6.6 23.6] | 7.3 [2.2 29.3] | 8.6 [4.7 23.3] | 18.5 [9.9 32.7] | 8.5 [3.0 23.4] | 9.7 [0.4 33.0] | 2.3 [-10.3 16.0] | 19.7 [8.8 27.0] | 13.4 [8.1 24.8] | 10.2 [1.7 29.7] | 13.2 [6.5 30.0] | 17.2 [10.1 34.7] |
|  | ssp370 | 17.6 [9.0 36.2] | 13.4 [3.1 38.6] | 13.7 [5.3 41.5] | 21.0 [12.8 45.9] | 13.8 [5.0 31.0] | 15.7 [0.7 40.7] | 2.8 [-3.6 11.9] | 25.3 [14.6 41.4] | 18.2 [10.2 39.4] | 17.4 [-5.0 34.1] | 19.6 [6.3 58.0] | 19.8 [13.2 44.8] |
|  | ssp585 | 24.9 [14.4 48.1] | 14.6 [7.3 47.0] | 25.4 [13.5 63.0] | 27.1 [16.6 61.6] | 22.8 [9.8 45.8] | 19.9 [5.3 54.1] | 9.2 [-1.0 29.9] | 35.2 [18.6 61.2] | 25.6 [14.2 50.0] | 24.2 [0.1 52.5] | 28.5 [13.9 78.7] | 23.9 [13.7 58.6] |

**Table E1.** Same as Table 2 but for relative anomalies only for snow cover and precipitation.



*Author contributions.* ML, MM and GK designed the study. ML produced the figures. ML and MM wrote the article and other authors contributed with suggested changes and comments. The ESA CCI snow product were provided by KN. All authors discussed the results and
provided critical feedback.

*Competing interests.* The authors declare that they have no conflict of interest.

*Acknowledgements.* We acknowledge the World Climate Research Programme (WCRP), which, through its Working Group on Coupled Modelling, coordinated and promoted CMIP6. We thank the climate modeling groups for producing and making available their model output, the Earth System Grid Federation (ESGF) for archiving the data and providing access, and the multiple funding agencies who support CMIP6
and ESGF. We thank the CLIMERI-France infrastructure (http://climeri-france.fr/) for making the CMIP6 data available. The authors are grateful for the ESA CCI+ initiative and the Snow project team that developed and made available the AVHRR GAC snow cover time series. This work is supported by the European Space Agency (ESA) Snow Climate Change Initiative (CCI) project (grant 4000124098/18/I-NB).





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
