# Peer review of "Climate change in the High Mountain Asia in CMIP6"

_Earth System Dynamics, 2021_

## Referee Comment (RC1)

**Review Comments**

The manuscript is based on a very important topic of the Climate Change in the High Mountain Asia using CMIP6 data for temperature, snow cover and precipitation. Different observed and reanalyzed data were used for the different parameters in the study to get the most appropriate picture. The authors here attempt to address performances of 26 CMIP6 GCMs over HMA for the historical period 1979-2014 and the future projections. 10 models were considered for the future projection and related analysis based on the availability of the 4 SSPs scenarios (SSP1-2.6, SSP2-4.5, SSP3-7.0, and SSP5-8.5) to estimate the anthropogenic emissions. The results may have some implications for the academic community. However, there are several areas where the manuscript can be improved. I have few comments that I am giving below:

**Minor comments**

1.  Use the term either "Snowpack" or "Snowcover" (L: 58).

2.  L: 89 sentence "Sections 6 ...." Should be restructured".

3.  Meaning of the sentence in L: 220 "Whenever possible......" is not clear. What are other datasets?

4.  A little more elaboration on the temperature is required in Section 3.2 to put light on the obtained results in the study.

5.  Sentence "For temperature, ERA-Interim......." (L: 252-254) is not in the flow of the preceding text and therefore need elimination from the manuscript.

6.  Use either "Sect." or "Section" in different paragraphs of the manuscript.

7.  Why is the in-depth discussion on APHRODITE missing in this Section 3.2 ?

8.  More description in Section 3.3 on the temperature, snow cover and precipitation is required to show the bias.

9.  Sentence "Correlation between temperature and snow cover........" (L: 300-303) should be part of Section 3.4.

10. Rather than temperature only, section heading of Section. 3.3 needs to reflect the snow cover as well as precipitation. Add more text in Section 3.3 for the discussion on precipitation.

11. Which test was adopted for the significance of the correlation and how was the level of significance selected to declare the value significant ?

12. Mention the threshold limit selected to declare the trend results significant in Section 4.1.

**13.** Comments on the figure and the tables have not been made at my end. Please add it!

---

## Author Comment (AC1)

**Reply RC1**

RC1: 'Comment on esd-2021-43', Anonymous Referee #1, 20 Jul 2021
https://doi.org/10.5194/esd-2021-43-RC1

**Climate change in the High Mountain Asia in CMIP6**

Mickaël Lalande et al. (https://esd.copernicus.org/preprints/esd-2021-43/#discussion)

**Review Comments**

We are grateful to referee #1 for having accepted to review this paper and for his helpful comments.

**Minor comments**

Comments of reviewers are displayed in *italic,* replies are in normal text and when additional text is added within a paragraph (between quotation marks), it is shown in **bold**.

1. *Use the term either "Snowpack" or "Snowcover" (L: 58).*
We replaced "snowpack" with "**snow cover**" to be consistent in this paper.

2. *L: 89 sentence "Sections 6 ...." Should be restructured".*
Replaced by: "**The results are discussed in Sect. 6 and the conclusions are presented in Sect. 7.**"

3. *Meaning of the sentence in L: 220 "Whenever possible......" is not clear. What are other datasets?*
Replaced by: "**To get confidence in the model bias quantification, we use further observational datasets, including GPCP precipitation, ESA CCI snow cover as well as ERA-Interim and ERA5 reanalysis.**"

4. *A little more elaboration on the temperature is required in Section 3.2 to put light on the obtained results in the study.*

"[...]. Indeed, the multimodel mean temperature is around 2 to 3~°C below the CRU observations in winter over HMA, while models and observations are much closer in summer (Fig.~\ref{fig:ac}a). These differences are more pronounced in the HK region (Fig.~\ref{fig:ac}b) with differences noticed both in winter (4 to 5~°C) and summer ($\sim$2~°C). **The cold bias appears in the multimodel mean (dark blue line) from October/November onwards, peaks between December and January, and then decreases until April/May, except in the HK area where the bias persists in summer. Nevertheless, the multimodel spread encompasses the observation and reanalyses datasets, suggesting a certain reliability of the CMIP6 models. This spread denoted with the confidence intervals at 50~\% and 90~\% of the multimodel ensemble (dark and light shadings)**

**highlight a higher dispersion between the models in winter than in summer, except for the HK region.**

> 5. *Sentence "For temperature, ERA-Interim......." (L: 252-254) is not in the flow of the preceding text and therefore need elimination from the manuscript.*

Ok, we removed the sentence.

> 6. *Use either "Sect." or "Section" in different paragraphs of the manuscript.*

"Section" was removed or replaced by "Sect." where it appeared in a middle of a sentence and was kept at the beginning of sentences as indicated in the manuscript composition instructions.

> 7. *Why is the in-depth discussion on APHRODITE missing in this Section 3.2 ?*

We did not want to drown the results in too many discussions, of which we already touch a little (L: 280-290) and that we complement in the discussion (Sect. 6, L: 581-590).

> 8. *More description in Section 3.3 on the temperature, snow cover and precipitation is required to show the bias.*

"The pattern of the temperature bias widely differs from one model to another (Fig.~\ref{fig:tas_bias}). However, most of the models show a cold bias, which is reflected by the multimodel mean reaching an average bias of $-$1.9 [$-$8.2 to 2.9]~°C. The cold bias **show common general features among the models, being** generally more pronounced at high elevation (Fig.~\ref{fig:clim}a)**, in particular over HK region as highlighted in Sect.~\ref{ssec:ac}**. The largest biases are found for the CNRM and IPSL models, with biases reaching almost $-$10~°C on average and exceeding locally $-$12~°C, especially over the western part of the TP and in the Karakoram area (HK region). Other models show slight positive or negative biases around $\pm$3~°C. **Some models show a positive bias at the edges of the plateau and over the Tien Shan (e.g., CESM2-FV2 and MIROC-ES2L) that contrasts with a cold bias on the Southern flank of the Himalaya. This is probably due to the low resolution of these models which does not allow to catch the atmospheric circulation over this high elevation narrow area (Fig.~\ref{fig:clim}b). The cold bias found in a large number of models is more pronounced in winter, a season during which it extends over almost the entire TP, whereas it is limited to the HK region in summer (not shown). Conversely, the warm bias found in some models is reduced in winter and exacerbated in summer.**

**As for temperature, the** snow cover **shows a general overestimation in the multimodel mean that extends homogeneously over the whole TP with slightly higher values northwest of TP and over HM (>30~%)** (Fig.~\ref{sfig:snc_bias})**. Surprisingly the multimodel mean shows a slight underestimation of snow cover of about 10% over the HK region, which seems contradictory with the intense cold bias pointed out simultaneously in this area. Indeed, the CRU dataset may overestimate temperature in this area due to a lack of observations, while the low resolution of the NOAA CDR simple binary product (grid cells with or without snow), might overestimate the snow cover in this often snowy area. The ESA CCI product shows a lower snow cover in general and in particular in this region (not shown). It is therefore possible that in the HK region the model biases actually reflect observation deficiencies, even if other factors affecting the model skill could be involved.** The annual multimodel mean of snow cover is overestimated by 12 [$-$13 to 43]~\% (or 52 [$-$53 to 183]~\% relatively) over HMA compared to NOAA CDR and can reach

locally an absolute difference of 40~\%, while a minority of models show a slight underestimation of snow cover (e.g. MPI-ESM1-2-HR, MPI-ESM1-2-LR, NorESM2-LM). **The annual overestimation of the snow cover in most models arises mainly from a too wide extension in the inner TP in winter (not shown). While the excess of snow melts in summer in most of the models, leading to a moderate bias during this season (Fig. 2), some models keep a persistent excess of snow even in summer (e.g. HadGEM3-GC31-LL, HadGEM3-GC31-MM and IPSL-CM6A-LR), which partly explains the large dispersion between the models in terms of annual biases.**"

For precipitation, see comment 10. for further discussion.

9. *Sentence "Correlation between temperature and snow cover........" (L: 300-303) should be part of Section 3.4.*

Sentence removed because it's already said in the first paragraph of Sect. 3.4.

10. *Rather than temperature only, section heading of Section. 3.3 needs to reflect the snow cover as well as precipitation. Add more text in Section 3.3 for the discussion on precipitation.*

3.3  -> **Spatial biases**

Additional discussion on precipitation:

"[...] The bias pattern **in terms of total precipitation** is **somehow proportional** to the climatological pattern **of precipitation**, with stronger biases in the Southeastern Himalaya, where high precipitation rates are observed **(Fig. B2). The quantification of the bias should be considered carefully for precipitation, because the APHRODITE data set strongly underestimates the precipitation rates at high elevations (Immerzeel et al., 2015). Anyway, the dry bias found in the southern flank of the Himalaya, coupled with a positive bias of precipitation over TP, suggests a too coarse resolution to represent the orographic barrier that blocks the Northward moisture flux, a limitation especially pronounced during the Asian summer monsoon that induces strong precipitation rates in the South of HMA.**"

11. *Which test was adopted for the significance of the correlation and how was the level of significance selected to declare the value significant ?*

We thank the reviewer for pointing out this point that was missing in the manuscript, here is the addition to the Sect. 2.5 (Numerical methods and computations):

L199 : "Trend computations are based on linear least-squares regression. We consider a 95~\% level of significance, corresponding to a p-value equal to 0.05, computed with a two-sided Wald test for which the null hypothesis corresponds to a slope equal to zero\footnote{\url{https://docs.scipy.org/doc/scipy/reference/generated/scipy.stats.linregress.html}}}.
The linear relationship between two datasets is estimated with the Pearson correlation coefficient. **We consider a 95~\% level of significance, corresponding to a p-value equal to 0.05, computed as follows: for a given sample with correlation coefficient $r$, the p-value is the probability that $\lvert r' \rvert$ of a random sample $x'$ and $y'$ drawn from the population with zero**

**correlation        would        be        greater        than        or        equal        to        $\lvert        r \rvert$.\footnote{\url{https://docs.scipy.org/doc/scipy/reference/generated/scipy.stats.pearsonr.html}} Note that the spatial correlation associated with p-values in Fig. 4 and Figures C1-3 does not include any dependency on the cell area. This arbitrary choice implies that the models are evaluated grid cell by grid cell and not per unit of surface. However, the impact on the spatial correlation is minor in our case, given that HMA is a relatively small area including model grid cells with areas that are relatively similar.**"

12. *Mention the threshold limit selected to declare the trend results significant in Section 4.1.*

L380: "Figure 6 shows a general positive trend for temperature in observations and models during both seasons. **Shading highlight the significant trends (p-value > 0.05), contours are used for not significant trends, and we consider that trends are robust when > 80 % of the models agree on its sign (hatching).**" (see RC2 for the addition of robustness)

13. *Comments on the figure and the tables have not been made at my end. Please add it!*

---

## Author Comment (AC2)

**Reply RC2**

RC2: 'Comment on esd-2021-43', Anonymous Referee #2, 25 Jul 2021
https://doi.org/10.5194/esd-2021-43-RC2

**Climate change in the High Mountain Asia in CMIP6**

Mickaël Lalande et al. (https://esd.copernicus.org/preprints/esd-2021-43/#discussion)

**Review Comments**

We thank referee #2 for having accepted to review this paper and for his constructive comments.

**Specific comments**

Comments of reviewers are displayed in *italic,* replies are in normal text and when additional text is added within a paragraph (between quotation marks), it is shown in **bold**.

1. *Introduction, L35: "As a large mid-tropospheric heat source…" The role of the Tibetan Plateau as a heat source for monsoon is a debated topic. Authors may refer to Boos and Kuang (2010; Dominant control of the South Asian monsoon by orographic insulation versus plateau heating. Nature 463, 218–222 (2010). https://doi.org/10.1038/nature08707)*

L35: "**The Asian summer monsoon provides almost 80~\% of the annual precipitation in the central and eastern parts of the Himalayas during the monsoon season (June-September) \citep{Bookhagen2010, Palazzi2013, Sabin2020}. Several studies suggested that the geographical configuration of the TP was enhancing the triggering of the Asian monsoon, this dry area acting as a heat source transferred to the mid-troposphere directly enhancing the vertical uplift typically found at the start of the summer monsoon** \citep{Li1996, Wu1998, Yihui2005, Wu2012}**. This finding has been partly questioned in other studies suggesting that the Himalayan chain is insulating the warm and moist air found over the Indian subcontinent from the cold areas found in TP \citep{Boos2010}. Therefore, the Himalaya itself and not the TP seems to be an essential geographical feature that favors vertical uplifts of warm and moist air masses, mainly on its southern flank.**"

2. *Taylor diagram is a useful tool in multi-model inter-comparison, especially when the models are compared with respect to the observations. I suggest that the authors use Taylor diagrams to compare temperature, precipitation, snow cover, etc. simulated by the models.*

The following Fig. X will be added in Section 3.5 (Metrics).

[Figure]

**Figure X.** Taylor diagram showing for the 26 models over HMA the 1979-2014 mean of the spatial pattern of temperature (a), precipitation (b) and snow cover (c). The observational reference is shown with a black star corresponding to CRU (temperature), APHRODITE (precipitation) and NOAA CDR (snow cover extent). ERA-Interim and ERA5 are shown with the black circles filled and non-filled respectively. The red pentagons correspond to the multimodel mean. The radial distance from the origin is proportional to the area-weighted standard deviation of the spatial pattern (normalized by the observation standard deviation). The area-weighted normalized centered RMSE between the model and the reference is proportional to the distance from the black star (light gray semi-circles). The area-weighted pattern correlation coefficient between the two fields is given by the azimuthal position.

After L360:

"The Taylor diagram (Taylor 2001) shown in Fig. X is used to investigate the realism of the spatial variability simulated in the models as compared to observational references. Overall, the models perform better for temperature than for precipitation, whereas the model skill is even

smaller for snow cover. The pattern correlation (PCC) ranges from 0.7 to 0.9 for temperature, whereas it takes lower values for precipitation varying from 0.6 to 0.8 for most of the models, except for HadGEM3-GC31-MM for which it reaches 0.9 and for 5 other models showing a lower PCC below 0.6. For snow cover, the model PCC is even lower and also heterogeneous among the models, varying from negative values (-0.17 for MIROC-ES2l) to a maximum of 0.8 (GFDL-CM4). Overall, the spatial variance is higher for almost all the models as compared to observations for both the temperature (the normalized standard deviation reaching 1.5 for the worst model) and the precipitation (the normalized standard deviation exceeding 4 for the worst model). This is the contrary for snow cover, a variable for which the models show smaller spatial heterogeneities in comparison to the observational reference, with a normalized standard deviation generally lower than 1, and varying between 0.4 and 1.4 for all the models. The larger temperature standard deviation found for the models is partly explained by the general cold bias over HMA that enhances the temperature contrast between the high elevation areas and the surrounding plains. The excess of precipitation found in the models over the area located under the influence of the Asian monsoons also explains the high standard deviation found in the models for this variable. In contrast, the low standard deviation found in the model for the snow cover is likely related to the too extended and too homogeneous snow cover over TP and its surrounding mountains, while the TP in observations is most often free of snow. Another interesting point is that both ERA-Interim and ERA5 do not perform much better than the CMIP6 models (except ERA-Interim for temperature and snow cover likely due to IMS snow cover assimilation over HMA), suggesting general weaknesses in the models used commonly for climate modeling and for the production of atmospheric reanalysis. While the multimodel mean is having intermediate performances among the models.”**

L361

“**Overall, it is challenging to discard any model from this spatial analysis, as well as RMSE and bias metrics, because of both a large heterogeneity of skill found among the models and a skill that varies also from one variable to another one for the same model.** This finding suggests [...]”

> 3. *Figure 6.: I suggest that the authors can try showing the agreement among the models in the sign of the trend in this figure. The hatching is often used to show statistically significant trends and not otherwise. Authors may try using contours for not significant trends, shading for significant trends, and hatching for points where > 60 % of the models agree on the sign of the trend.*

Here is the modified version of Fig. 6 taking into account the suggestions of the comment 3. The threshold for the points where the models agree on the sign of the trend is set to 80 % in order to show the robustness of the trends (because 60% showed too many hatched areas including quite a lot of areas where the multimodel mean trends were non-significant).

[Figure]

Figure 6. DJFMA (left) and JJAS (right) trends computed over 1979-2014 for temperature (a-d), snow cover (e-h) and precipitation (i-l). CRU temperature, NOAA CDR snow cover and APHRODITE precipitation observations trends (DJFMA: a, e, i and JJAS: c, g, k) are compared to the multimodel mean computed with the first realization for each model (DJFMA: b, f, j and JJAS: d, h, l). **Contours are used for not significant trends, shading for significant trends (p-value < 0.05), and hatching for points where > 80 % of the models agree on the sign of the trend.**

We will similarly modify Figures D3-5 and adapt the text in Sect. 4.1 (Trends) to reflect these changes.

4.    *Figure 9: "2081-2010" in the caption should be 2081-2100".*
Thanks for noticing that, we corrected it!